# Genomic analyses of a livestock pest, the New World screwworm, find potential targets for genetic control programs

Maxwell J. Scott [1✉], Joshua B. Benoit[2], Rebecca J. Davis[1], Samuel T. Bailey[2], Virag Varga [2], Ellen O. Martinson [3], Paul V. Hickner[4], Zainulabeuddin Syed[4], Gisele A. Cardoso[5], Tatiana T. Torres [5], Matthew T. Weirauch [6,7,8], Elizabeth H. Scholl[9], Adam M. Phillippy [10], Agustin Sagel[11], Mario Vasquez[11], Gladys Quintero[11] & Steven R. Skoda[12]

The New World Screwworm fly, *Cochliomyia hominivorax*, is a major pest of livestock in South America and Caribbean. However, few genomic resources have been available for this species. A genome of 534 Mb was assembled from long read PacBio DNA sequencing of DNA from a highly inbred strain. Analysis of molecular evolution identified 40 genes that are likely under positive selection. Developmental RNA-seq analysis identified specific genes associated with each stage. We identify and analyze the expression of genes that are likely important for host-seeking behavior (chemosensory), development of larvae in open wounds in warm-blooded animals (heat shock protein, immune response) and for building transgenic strains for genetic control programs including gene drive (sex determination, germline). This study will underpin future experiments aimed at understanding the parasitic lifestyle of the screwworm fly and greatly facilitate future development of strains for efficient systems for genetic control of screwworm.

[1] Department of Entomology and Plant Pathology, North Carolina State University, Campus Box 7613, Raleigh, NC 27695-7613, USA. [2] Department of Biological Sciences, University of Cincinnati, McMicken School of Arts and Sciences, Cincinnati, OH 45221, USA. [3] Department of Biology, University of New Mexico, Albuquerque, NM 87131, USA. [4] Department of Entomology, University of Kentucky, Lexington, KY 40546, USA. [5] Department of Genetics and Evolutionary Biology, University of São Paulo, São Paulo, Brazil. [6] Center for Autoimmune Genomics and Etiology, Cincinnati Children's Hospital Medical Center, Cincinnati, OH 45229, USA. [7] Divisions of Biomedical Informatics and Developmental Biology, Cincinnati Children's Hospital Medical Center, Cincinnati, OH 45229, USA. [8] Department of Pediatrics, University of Cincinnati College of Medicine, Cincinnati, OH 45267, USA. [9] Bioinformatics Research Center, North Carolina State University, Campus Box 7566, Raleigh, NC 27695-7566, USA. [10] Genome Informatics Section, Computational and Statistical Genomics Branch, National Human Genome Research Institute, Bethesda, MD 20892, USA. [11] USDA-ARS, Screwworm Research Unit, Pacora, Panama. [12] USDA-ARS, Tick and Biting Fly Research Unit, Knipling-Bushland Livestock Insects Research Laboratory, 2700 Fredericksburg Rd., Kerrville, TX 78028, USA. ✉email: mjscott3@ncsu.edu

The New World Screwworm, *Cochliomyia hominivorax*, is a blow fly that is an obligate parasite of warm-blooded animals in tropical and subtropical regions of South America and the Caribbean[1]. Females seek animals and lay their eggs on the skin, often near open wounds. Larvae feed on the animal's tissues, enlarging the wound, which can cause death if not treated[2]. The common name for the species comes from the larval behavior of burrowing into the host's tissues. After cessation of feeding, larvae leave the animal and pupate in the soil. The life cycle takes ~3 weeks.

Economic losses due to screwworm infestation of livestock are significant. In 2005 it was estimated that in South America alone annual losses were ~$US 3.6 billion[3]. Due to its economic importance, screwworm was eradicated from North and Central America in perhaps the most successful application of the sterile insect technique or SIT[4]. SIT involves mass rearing of the insect, sterilization by ionizing radiation, and repeated releases over the targeted area. The current mass rearing facility in Pacora, Panama, produces ~25 million sterile flies per week. The flies are released daily along the Colombian border to prevent reinfestation from South America. As SIT is more efficient if only males are released, considerable effort has been made to make conditional female lethal strains. Several tetracycline-repressible female pupal lethal strains were made and evaluated for characteristics important for mass rearing and performance in the field[5]. Since larval diet is a major cost for the mass rearing facility, it is advantageous if females die at the embryo stage. Consequently, we developed two component transgenic embryo sexing strains (TESS) of the related blowfly *Lucilia cuprina*[6,7]. Building these strains required identification of gene promoters active in the early embryo as well as sex determination and proapoptosis genes. For suppression of the targeted pest population, the TESS would be mass-reared and males released at regular intervals in excess of the wild population. As such a program could be costly, several alternative genetic strategies are under development that potentially could be more efficient and thus less expensive[8]. For example, strains with Cas9-based homing gene drives targeting genes required for female development or fertility could potentially be much more efficient than male-only SIT[8,9]. In addition to identifying suitable target genes, building gene drive strains requires identification of suitable gene promoters for driving Cas9 expression in the germline and for expression of gRNAs.

A better understanding of *C. hominivorax* development and behavior would facilitate genetic control programs. Female host-seeking behavior is of interest for improving our understanding of the biology of screwworm and for development of improved traps for monitoring and possible control. Gravid screwworm females are attracted to odors from screwworm-infested wounds[10]. The odors appear to be produced by bacteria in the wounds and from screwworm larvae[10,11]. Some of the odors produced by bacteria that attract flies are also used for swarming[12]. That is, blow flies are responding to quorum sensing chemicals used by bacteria for cell–cell communication. The traps currently used for screwworm detection by the eradication program use *Swormlure-4* as the attractant[13]. *Swormlure-4* is a mix of 10 chemicals including dimethyl disulfide, benzoic acid, indole, and phenol. The latter is also a bacterial swarming signal[12]. A complete repertoire of chemosensory genes could allow for the identification of more effective chemical control.

In the mass rearing facility, screwworm embryos complete embryo development in 6–7 h at 39 °C[5]. The temperature was chosen to closely match the body temperature of cattle. It is not known if under these conditions screwworm embryos and larvae experience thermal stress but the temperatures are higher than is optimal for the black soldier fly[14], a fly which is also found in tropical and subtropical regions of the Americas. In addition to

high temperature, in the hostile wound environment screwworm larvae would need to respond to bacteria growing in the wound and possibly also the host immune system. Thus, genes that play a role in the *C. hominivorax* immune system are of interest in understanding its parasitic lifestyle.

*C. hominivorax* has five pairs of metacentric autosomes of approximately equal size and a smaller pair of X and Y sex chromosomes[15]. The genome size of the J06 strain was determined by flow cytometry to be 441.5 Mb for female and 443.8 Mb for male[16]. J06 is the current strain under mass production in Panama. The strain was established by interbreeding 12 isofemale lines that were established from flies caught in Jamaica in 2006[1]. With the long-term aims of improving our understanding of screwworm biology and facilitating the further development of strains for genetic control, we have assembled and annotated the genome of the J06 strain, performed community annotation for genes categories of interest, performed selection analyses of predicted genes, and examined gene expression at different stages of development and between sexes.

## Results

**Genome assembly and gene prediction.** The J06 strain was inbred for 10 generations of full-sibling single-pair matings to reduce heterozygosity. High molecular weight genomic DNA (>200 kb) was prepared from late stage (6 h) embryos from the inbred stain. We found this stage of development consistently produced higher quality DNA than DNA prepared from adults, which could be due to the presence of the exoskeleton causing shearing during isolation[17]. In all, 20 and 30 kb libraries were prepared for PacBio DNA sequencing. A total of 39255 Mb of sequence was obtained from the long reads (8767–10044 bp) which represents 89-fold coverage given a genome size of 440 Mb. The genome was assembled using Canu with default parameters modified for residual heterozygosity in the inbred strain (Table 1).

The assembly size of 534 Mb is larger than expected and could indicate that some heterozygous alleles were assembled onto separate contigs. To assess the completeness of the assembly, we searched for the presence of 2799 core Diptera genes utilizing BUSCO[18]. The results suggest a very complete set (94.1% complete and single copy) with little fragmentation (1.6%). The level of duplication (3.6%) was also low and suggests that, at least for conserved protein-coding genes, few single-copy genes have been placed onto separate contigs due to any residual heterozygosity. Nineteen of the 2799 genes were missing. The BUSCO scores are similar to that for other fly genomes, indicating the *C. hominivorax* draft genome is of comparable quality (Fig. 1). A set of protein-coding genes was obtained by first mapping RNAseq reads from different stages (see below) to the reference genome, which was then used as input for Braker to create a training set for Augustus (Supplementary Data 1). Of the 22,491 predicted protein-coding genes, 20,975 have a match to a protein in the NCBI NR database (Supplementary Data 2).

The *C. hominivorax* genome was searched for repetitive DNA sequences including transposable elements. Repetitive sequences were identified by running Repeatmasker using a de novo repeat

**Table 1 Summary of assembly statistics.**

| | |
|---|---|
| Contig count | 3663 |
| Average length | 145805.3 bp |
| N50 | 616429 bp |
| Total length | 534.1 Mb |
| GC content | 27.7% |

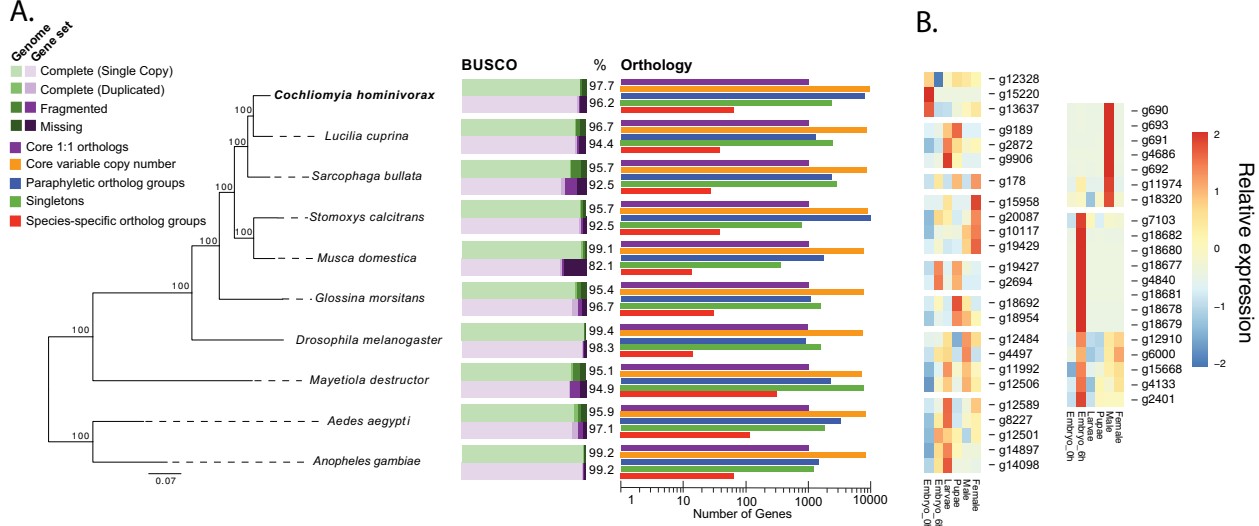

**Fig. 1 Phylogenetic placement and genomic comparisons for *C. hominivorax* and other fly species. a** The phylogenetic analysis places *C. hominivorax* as a sister species to the Australian sheep blow fly, *Lucilia cuprina*. The phylogeny is built using RAxML and it is based on amino acid sequences from 612 single-copy genes that are present in all 10 species. Bootstrap values are shown for every node. Benchmarking Universal Single-Copy Orthologs[92] analyses were based on the dipteran dataset associated with this application (odb8). Green, genome; Purple, predicted gene sets. Orthology-based analyses of protein-coding genes between eight fly species determined the number of genes in single-copy core clusters, variable-copy number core clusters, paraphyletic clusters (non-core, non-species specific), singleton, and species-specific clusters, based on OrthoFinder[69]. **b** expression profiles for genes unique to *C. hominivorax* across development based on Supplementary files 8–13. Each expression value represents the average of three biological replicates for each stage. Relative expression levels are compared across the rows.

library that was made by running RepeatModeler2. Overall, 45.2% of the genome was repetitive with most repeats listed as unknown (25.3% genome). The unknown repeats were typically in the size range of 50–350 bp, repeated hundreds to thousands of times and were detected on most of the contigs. Of the transposable elements, the most abundant were DNA elements (6.9% of the genome) and LINEs (2.3% genome; Supplementary Data 3). *Tc1/mariner-Tc1* and *hAT-Ac* were the most common DNA transposons accounting for 5.37 and 0.67% of the genome, respectively. Similarly, in the medfly *Ceratitis capitata*, the majority of DNA elements were from the *Tc1/mariner* super-family[19]. *Jockey* (0.76% genome) was the most abundant non-LTR retrotransposon and *Pao* the most common LTR transposon (0.39% genome). Helitrons comprise 6.27% of the genome.

**Comparative genomic analyses**. The phylogenetic relationship of *C. hominivorax* to eight other Diptera was examined using a set of 612 single-copy genes present in all genomes. This analysis places *C. hominivorax* as most closely related to the Australian sheep blowfly, *Lucilia cuprina*, followed by the flesh fly, *Sarcophaga bullata*, as expected[20] (Fig. 1). Orthology comparison among these fly proteomes revealed *C. hominivorax*-specific gene sets, which are predominantly expressed in the embryo and male stages. The male-specific gene set has a high enrichment for secreted peptides, suggesting a putative role as a component of the male ejaculate.

**Predicted chemosensory genes**. We identified 78 canonical olfactory receptors (ORs) plus the olfactory receptor co-receptor (*Orco*) in the *C. hominivorax* genome, which is similar in number to *M. domestica* and *S. calcitrans* (Table 2, Supplementary Data 4, and Supplementary Fig. 1). Despite minor differences in the size of some gene lineages due to apparent birth-and-death evolution, most lineages were maintained with at least one gene copy resulting in OR repertoires of similar size and content (Supplementary Data 4 and Supplementary Fig. 1). Seventy-seven

**Table 2 The number of odorant, gustatory, and ionotropic receptors (ORs, GRs, and IRs), and odorant binding proteins (OBPs) in the screwworm fly (*C. hominivorax*) compared with other Diptera.**

| Species | ORs | GRs | IRs | OBPs |
|---|---|---|---|---|
| *C. hominivorax*[a] | 79 | 84 (77) | 88 (83) | 51 |
| *S. calcitrans*[b] | 73 | 113 (73) | 145 | 90 |
| *M. domestica*[c] | 85 (84) | 100 (76) | 110 | 93 |
| *D. melanogaster*[d] | 62 (60) | 68 (60) | 65 | 52 |

The numbers reported are the total number of encoded proteins, while the numbers in parentheses are the number of genes where splice variants were predicted.
[a]Annotated here.
[b]Olafson et al.[80].
[c]Scott et al.[78].
[d]FlyBase (FB2019_02, released April 2019).

gustatory receptor (GR) genes encoding 84 proteins were predicted in *C. hominivorax* (Table 2, Supplementary Data 4, and Supplementary Fig. 2). The size of the ionotropic receptor (IR) repertoire is smaller than the muscids, with 83 IR genes encoding 88 IRs in *C. hominivorax*, and 110 and 145 encoded IRs in *M. domestica* and *S. calcitrans*, respectively (Table 2, Supplementary Data 4, and Supplementary Fig. 3). The larger number of IRs in the muscids is due primarily to a single, highly expanded lineage neighboring *ChomIr137* and *ChomIr138* (Supplementary Data 4 and Supplementary Fig. 3). The OBP gene family is comprised of 51 OBPs, which is considerably smaller than the muscids, which have 90–93 OBPs (Table 2, Supplementary Data 4, and Supplementary Fig. 4). In summary, the size and content of the *C. hominivorax* OR repertoire is similar to *M. domestica* and *S. calcitrans*. However, there are fewer encoded GRs, IRs, and OBPs in *C. hominivorax* compared to the muscids.

**Developmental RNAseq analyses**. We performed developmental RNA-seq analyses based on methods used for other dipterans[21].

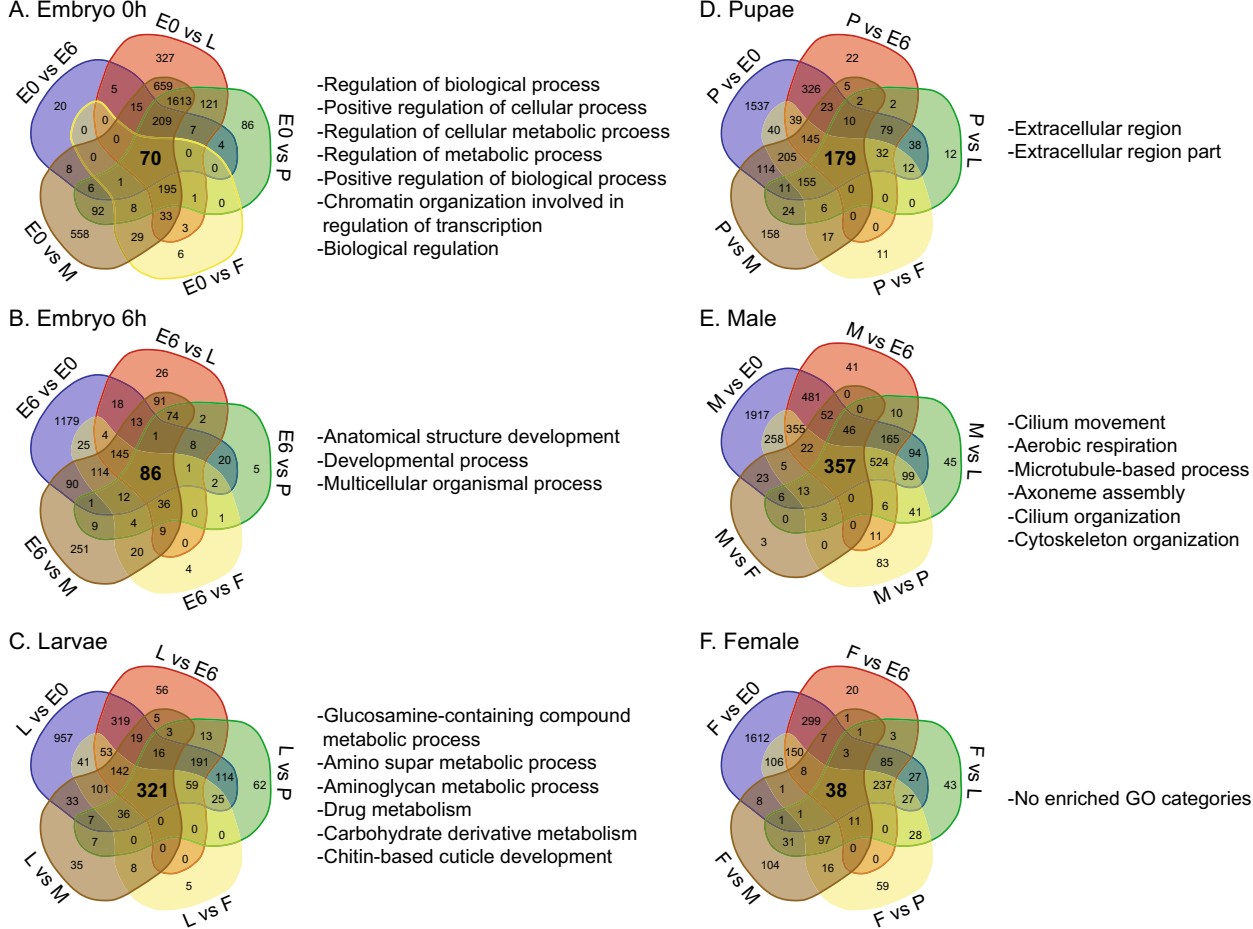

**Fig. 2 Genes uniquely enriched in the screwworm throughout development and associated gene ontology enrichment. a** Genes enriched in embryo 0–1 h after egg deposition (left) and gene ontology (right). **b** Genes enriched in embryo 6 h after egg deposition (left) and gene ontology (right). **c** Genes enriched in larvae (left) and gene ontology (right). **d** Genes enriched in pupae (left) and gene ontology (right). **e** Genes enriched in males (left) and gene ontology (right). **f** Genes enriched in females (left) and gene ontology (right). Genes are denoted as enriched within a specific stage if expression is significantly higher compared to all other stages. Gene ontology for enriched sets were determined with the use of gProfiler and visualized with REVIGO.

Our goal was to establish specific gene sets enriched within each developmental stage (embryos 0–1 h after oviposition, embryos 6 h after oviposition, larvae, pupae, males, and females). Two methods were used, which consisted of pairwise comparisons among all groups and weighted gene co-expression network analysis (WGCNA)[22]. WGCNA identifies genes with co-expression patterns between samples.

Pairwise comparison revealed distinct gene sets for each developmental stage (Fig. 2, Supplementary Data 5). For embryos at 0–1 h and 6–7 h after oviposition, sets of 70 and 86 genes were identified as enriched, respectively. At 39 °C, *C. hominivorax* takes ~7 h to complete embryogenesis. The embryo gene sets showed considerable enrichment for developmental processes likely associated with embryogenesis and early larval development (Fig. 2). Larval enriched gene sets were associated with cuticle development and aminoglycan metabolic processes, which play important roles during the rapid larval growth. The pupal period showed enrichment for aspects associated with extracellular region. The 357 genes enriched in males were associated with aspects that underlie sperm generation, such as cilium organization and axoneme assembly. The female gene set includes vitellogenin and vitellogenin receptor, which are critical for oogenesis. As transcription factors (TFs) are critical to many biological aspects, we conducted a large-scale identification and expression analyses of these genes (Fig. 3 and Supplementary

Data 6 and 7). The TFs for *C. hominivorax* were comparable to that observed for other flies, with the largest families being $C_2H_2$ zinc finger and homeodomain. Expression of the TFs across development showed that most TFs have the highest transcript levels in embryos and males (Fig. 3 and Supplementary Data 6). These specific TFs are likely to play critical roles during development or in specific sexes.

WGCNA identified specific modules of genes with similar expression for each developmental stage and sex (Fig. 4 and Supplementary Data 8). GO analysis of genes in stage-specific modules revealed a wide range of GO categories for pupae and males, which were attributed with 49 and 44 GO categories, respectively (Fig. 4c). Male modules showed enrichment in GO categories involved in production of sperm. Pupae modules showed enrichment in numerous development-related GO categories as well as cell signaling and adhesion. Larvae and females had fewer enriched categories based on WGCNA. Embryonic and larval stage modules showed enrichment in GO categories attributed to metabolism and growth. While there was overlap, WGCNA identified many unique GO categories that were not identified in the pairwise comparison of expression.

**Genes under positive selection in *C. hominivorax* and their developmental expression.** To identify genes that may be under

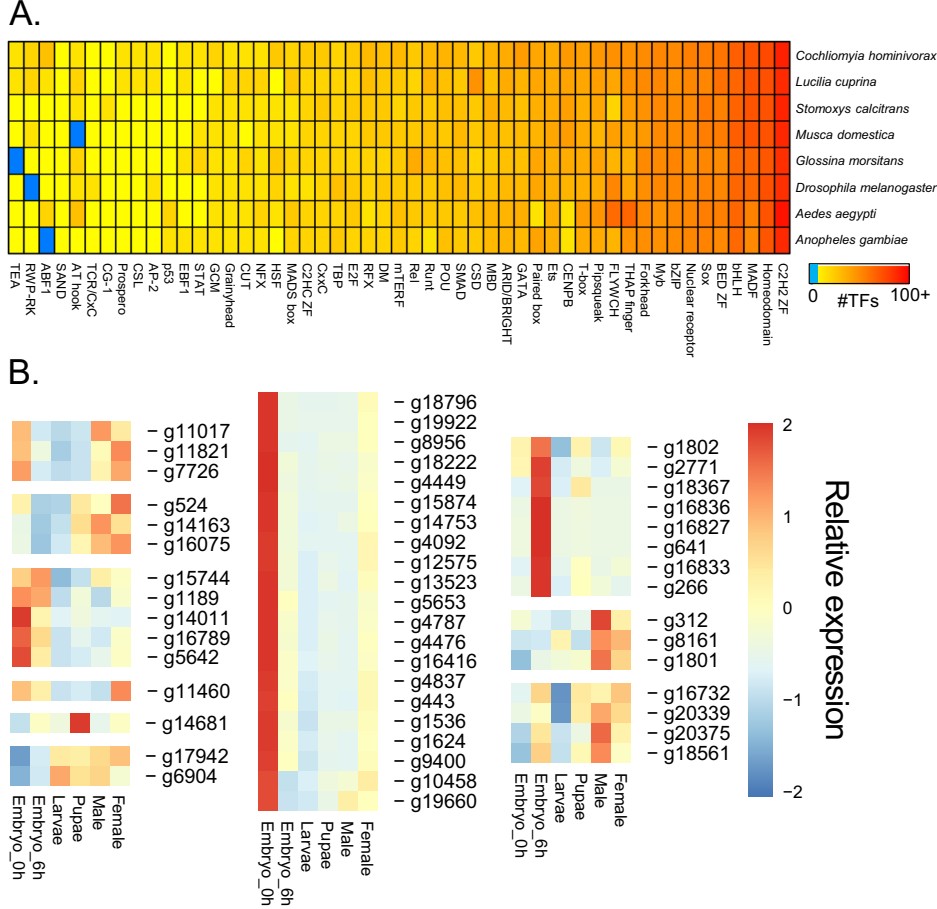

**Fig. 3 Distribution and expression of transcription factor families for specific dipterans. a** Heatmap depicting the abundance of transcription factor (TF) families across a collection of insect genomes. Each entry indicates the number of TF genes for the given family in the given genome, based on presence of DNA-binding domains. Color key is depicted at the top (light blue means the TF family is completely absent) – note log (base 2) scale. **b** Expression values at different stages of development for transcription factors that have significantly higher expression within one developmental stage based on Supplementary Data 6. Blocks represent genes with similar expression patterns across development.

positive selection in *C. hominivorax*, we recovered a set of 4489 orthologous genes between *C. hominivorax* and other dipterans (see Methods section). After amino acid alignment, back translation and masking, 14 genes were removed due to poor alignment; 4475 were retained for the molecular evolution analysis. We looked for evidence of positive selection on ortholog genes in the 12 tested species and in the *C. hominivorax* branch. After filtering genes by dS and dN/dS ratio (ω), all orthologs were kept in the model 1 (free ratio model), and 4239 in model 2 (two-rates model). In both models, almost all genes exhibited evidence of strong purifying selection (dN/dS < 1).

We compared the distribution of the dN/dS ratios across categories of genes by using GO terms. The distribution of dN/dS ratios for the genes within each term was also compared to the genome-wide distribution (Fig. 5). All classes of genes retained for this analysis were strongly constrained; the highest median estimate of dN/dS was 0.04 for six terms ("transport", "phosphorylation", "extracellular space", "serine-type endopeptidase activity", "G-protein coupled receptor activity", and "kinase activity"). Comparing the distributions of the ratio across genes within each GO term and the genome, nine terms had distributions that were significantly different from the genome-wide distributions. Four of them had genes more constrained than the other genes of the genome (Mann–Whitney tests, with alpha set to 0.05, and false discovery rate correction for multiple tests): three in the "Biological Process" category ("small GTPase

mediated signal transduction", "negative regulation of transcription from RNA polymerase II promoter", and "regulation of transcription from RNA polymerase II promoter"); two in the "Molecular Function" category ("chromatin binding" and "sequence-specific DNA binding transcription factor activity"); and one in the "Cellular Component" category ("nucleus"). Genes binned in these categories are, in general, critical to activities necessary for the maintenance of the cell cycle, and therefore are under strong purifying selection. Some of these classes have a shared constraint across distantly related phyla. The "small GTPase mediated signal transduction" class, for instance, has also shown a similar pattern of purifying selection in very divergent taxa such as nematodes, mammals, and plants[23,24]. The classes with accelerated evolution compared to the other genes in the *C. hominivorax* genome were "phosphorylation", "extracellular space", and "integral to membrane".

Of the genes associated with extracellular space, most have expression across all samples (Fig. 5). A few have enriched expression (fourfold higher) within a specific developmental stage, which have similar expression profiles to orthologs in *Drosophila*[25]. For phosphorylation, there may be a more direct link to biological significance (Fig. 5). As an example, genes with enriched male expression have orthologs in *Drosophila* with increased expression in either the testes, male accessory gland, or both. Specifically, creatine kinase is highly expressed in the testes of both *C. hominivorax* and *Drosophila*[25], highlighting the

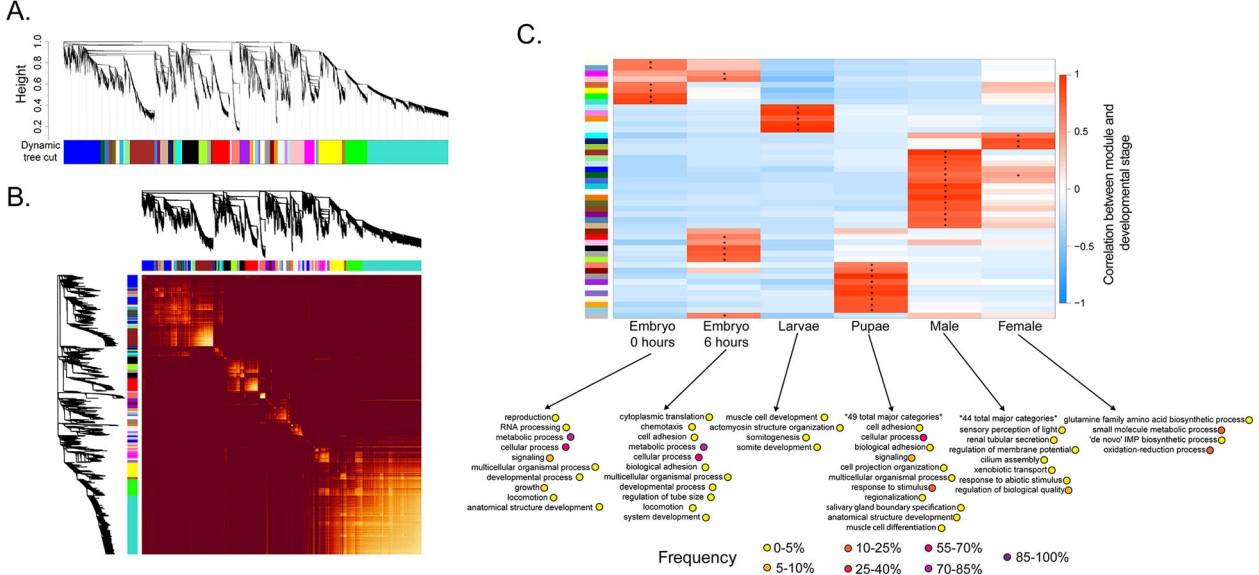

**Fig. 4 Weighted gene co-expression network analysis (WGCNA) across developmental stages for the screwworm, *C. hominivorax*. a** Average linkage hierarchical clustering dendrogram of the genes. The expression distance is shown in the *y*-axis. Modules, designated by color code in the *x*-axis, are branches of the clustering tree. These modules are made of genes with a high correlation of their expression levels across the developmental stages. **b** Relationship between each module based on similarity of expression between all genes for each developmental stage. The light yellow color show high topology overlap (high interconnection), and the dark color represents low overlap (high interconnection). **c** Correlation of gene expression patterns for each module eigengenes in relation to developmental stages. Each row corresponds to a module eigengene and columns are developmental stage. Asterisk represents values with a significant positive correlation for based on a regression-based *p*-value for assessing the statistical significance (*P* < 0.05) between a stage and module[91]. Some modules are correlated with specific developmental stages. Gene ontology (GO) analysis of eigengenes associated with each stage was conducted with g:Prolifer[89]. Frequencies represent the relative number associated with the specific GO compared to the GOA database[93].

importance of this factor in testes function. These combined selection-based GO categories followed by RNA-seq analyses reveal potential genes of interest for *C. hominivorax*, specifically those involved in reproduction that are shared, but evolving, among higher flies.

Forty genes are likely to be under positive selection in the *C. hominivorax* branch (Fig. 6 and Supplementary Data 9). The *D. melanogaster* orthologs of 19 of these genes have no annotated known function. This pattern was also observed in the comparison of the genomes of 12 *Drosophila* species[26]. The *Drosophila* orthologs of the remaining 21 genes are involved in diverse functions and molecular processes such as regulation of transcription (*lmd*, *Nf-YC*, and *gcm2*), mRNA splicing (*barc*), translation (*epsilonCOP*), signaling (*Dh31-R*), transport (*slo*, *AP-2sigma*), tracheal development (*crim*), development of fly organs (*Ror*, *l(1)sc*, *cv-2*), diverse mitochondrial activities (*mRpL4*, *mtSSB*), sensory perception of smell (*Obp8a*), regulation of circadian cycle (*Nca*), and some pleiotropic genes with many reported functions (*Vps25*, *S6k*, *Jra*, *CycJ*, and *robo1*)[27].

Of these 21 genes, four are of particular interest: *ChDh31-R* ($\omega = 9.87$, likelihood ratio test, *p*-value $= 1.77\mathrm{e}^{-94}$), *ChS6k* ($\omega = 3.19$, likelihood ratio test, *p*-value $= 2.03\mathrm{e}^{-112}$), *Chrobo1* ($\omega = 1.24$, likelihood ratio test, *p*-value $= 1.94\mathrm{e}^{-72}$), and *Chcrim* ($\omega = 7.27$, likelihood ratio test, *p*-value $= 1.61\mathrm{e}^{-227}$). *Drosophila* flies exhibit a daily cycle of temperature preference; during daytime they prefer higher temperatures, while lower temperatures are preferred at night. This temperature preference rhythm is mediated by the neuropeptide diuretic hormone 31 receptor (Dh31-R) in *D. melanogaster* during its active phase[28]. Unlike what is found in *Drosophila*, *C. hominivorax* larvae are at a constant temperature in the wounds of a living host. Perhaps this gene played a role in adaptation of the species to the body temperature of the host. It will be of interest to determine if

ChDh31-R mediates the preference of females for warm-blooded vertebrates. Among many other functions, the ribosomal protein S6 kinase modulates hunger response by insulin-like and neuropeptide Y-like signaling pathways in *D. melanogaster* larvae[29]. The up-regulation of the *S6k* gene reduces feeding rate and foraging in starved larvae, while its down-regulation triggers these same behaviors. Studying larval feeding behavior, and the underlying genes and pathways in *C. hominivorax* is key to understanding the origins and evolution of parasitism in Calliphoridae. The *Drosophila roundabout1* (*robo1*) gene is regulated by *fruitless* and plays an important role in male mating behavior[30]. Finally, *Drosophila crimpled* (*crim*) is one of four Ly6-like proteins required for septate junction formation[31]. RNAi knockdown of *crim* in trachea caused tube size defects[31]. A distinguishing feature of *C. hominivorax* are the thick and dark larval trachea tubes compared to its close relative *C. macellaria*[32].

Expression analysis of the genes under positive selection revealed distinct expression profiles (Fig. 6b). A majority of the genes under selection showed the highest expression levels within the embryos, either at 0 or 6 h of development. This was followed by those expressed highly within males (Fig. 6b). The *Chslowpoke* (*Chslo*) and *ChDh31-R*, discussed previously, are most highly expressed in males and females. Five of the other genes with high male expression have a sex-specific enrichment for *Drosophila*[25]. Indeed, orthologs for all five of these genes are expressed highly within the testes or accessory glands of male *Drosophila*[25]. These include CG8292, CG3698, CG3687, and CG10947. One of the five genes, *ChObp8a* (g14187), is of interest as the expression of odorant-binding proteins in male accessory gland was also previously observed in the tsetse fly[33]. Thus, the ChOBP8 protein could play an important role in accessory gland function in *C. hominivorax*. Of those with the highest expression in the embryos, most are also expressed at least at moderate levels in

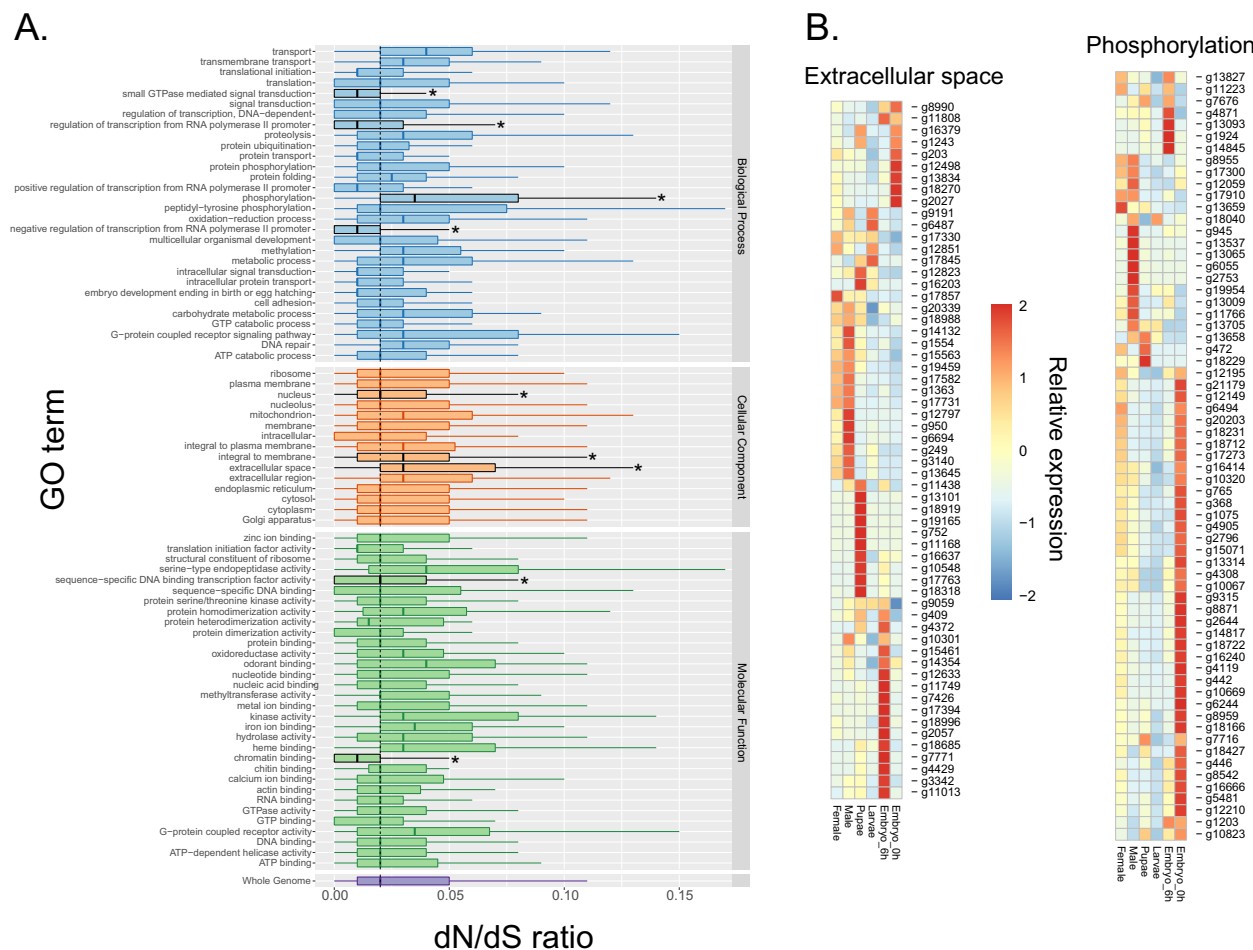

**Fig. 5 Gene ontology categorizes under selection in *C. hominivorax*. a** Distribution of the d$N$/d$S$ ratios across GO terms. GO terms represented by at least 30 genes were used. Some categories have accelerated rates of evolution compared to the other genes of the genome (median d$N$/d$S$ of the genes in the category is higher than the median of the other genes in the genome). Other categories are more constrained than the genomic average. The center lines in the boxes indicate the median of each distribution. Lower and upper hinges correspond to the first and third quartiles, and the whiskers extend to the largest or smallest value, no further than 1.5x the inter-quartile range. Outliers are not shown. The dashed vertical line shows the genome-wide median. The asterisks show the terms with d$N$/d$S$ distributions statistically different from the genome-wide distribution (Mann–Whitney $U$ test). **b** Developmental RNA-seq of genes belonging to two gene ontology categories with median d$N$/d$S$ higher than the genome, based on Supplementary Data 8. Each expression value represents the average of three biological replicates for each stage.

other stages (Fig. 6b and Supplementary Data 5), suggesting these genes likely have functions in most developmental stages.

**Heat shock protein (*hsp*) genes.** Twenty-two heat shock proteins were identified in the *C. hominivorax* genome, 17 of which were identified via the gene prediction and annotation process and an additional five via Blast search (Supplementary Data 6). Of the 22 putative HSPs, a Pfam[34] search indicates five are members of the RHOD (Rhodanese Homology Domain) superfamily and one a member of the GroEL family. The remainder are in the HSP20/alpha crystalline, HSP70, and HSP90 families. Most of the predicted *hsp* genes are found in two clusters. The five *hsp* genes that show homology to *Drosophila hsp68* and *hsp70* are found in one cluster (Supplementary Fig. 5). Nine of the genes that encode small HSPs similar to *Drosophila* HSP23 or HSP27 proteins are found in a second cluster (Supplementary Fig. 5). The remaining genes, orthologs of *Drosophila hsp60* (g16758), *hsp83* (g13430), and additional small *hsp* genes are single-copy genes found in different contigs.

We next examined the developmental expression profiles of the *hsp* genes. The *hsp68*/*hsp70* genes show low expression in 0–1 h embryos and adults (Fig. 7 and Supplementary Fig. 6). However,

6 h embryos, larvae and pupae show 10–35 times higher expression than 0–1 h embryos. In all, 6 h embryos and larvae were reared at 39 °C, pupae at 31 °C, and adults and 0–1 h embryos at 25 °C. Thus, it appears *C. hominivorax* may be responding to the higher temperatures used for rearing the pre-adult stages. The highest levels of expression of the *hsp68*/*hsp70* genes are seen in pupae. The small *hsp* protein genes show distinct expression profiles but most fall into one of two patterns. One group are strongly expressed in adult females and in 0–1 h embryos, suggesting high maternal expression. Expression levels are not as high at other stages. The second group show low expression in 0–1 h embryos but much higher expression by 6 h of development. These *hsp* genes are not expressed at higher levels in females than males. Not all small *hsps* fit into these two patterns. For example, g15959 is expressed at much higher levels in pupae than other stages.

**Immune response genes.** We searched for orthologs of 232 *Drosophila melanogaster* genes (some with multiple isoforms) that are part of the immune response and function in seven different pathways. These represent 218 distinct genes, of which 149 had at least one predicted screwworm protein, based on the

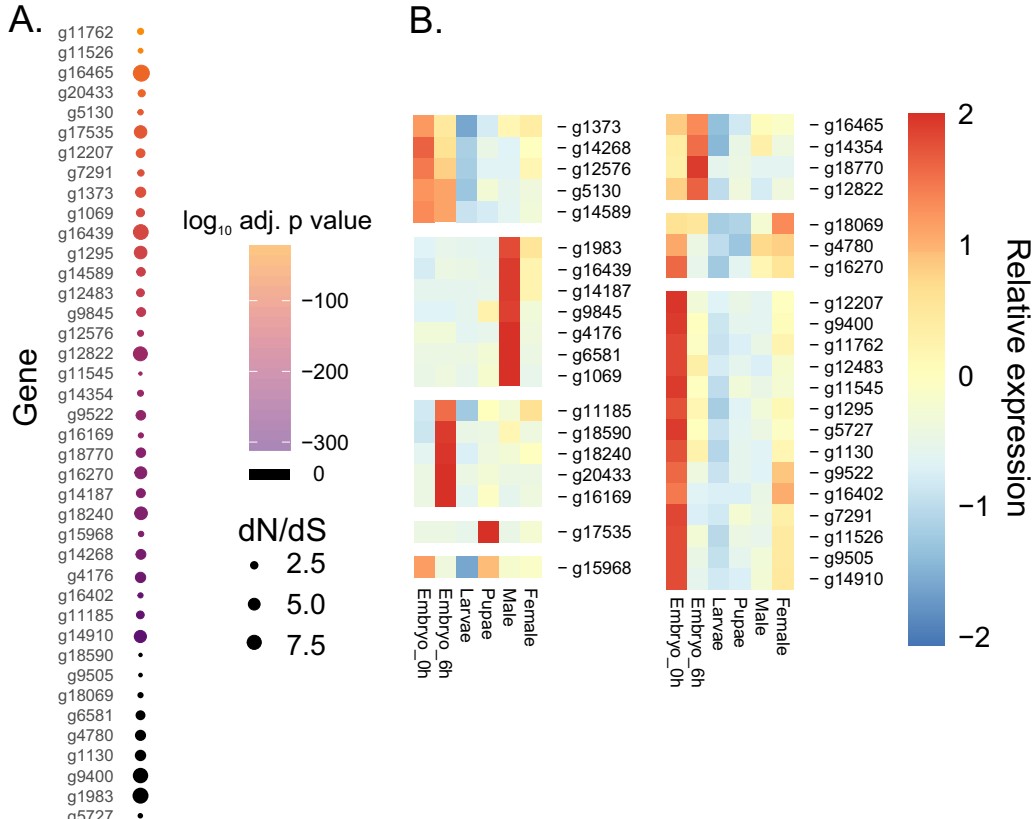

**Fig. 6 Genes under positive selection in *C. hominivorax* and their developmental expression. a** Forty genes with evidence of positive selection. Dot size indicates d*N*/d*S* ratio. The larger the dot, the higher the positive selection onto the protein (averaged over all codons). The color indicates the level of significance in the comparison of the two-ratio model with the null hypothesis of neutral evolution; it shows that the null hypothesis of neutral evolution was rejected for these genes. Adjusted *p*-values are log transformed. 0, indicates a *P* value of 0, which cannot be plotted on a log scale. **b** Expression profiles of the genes evolving under positive selection in six developmental stages, based on Supplementary Data 8. Each expression value represents the average of three biological replicates for each stage. Blocks are based upon similarity in expression among developmental stages.

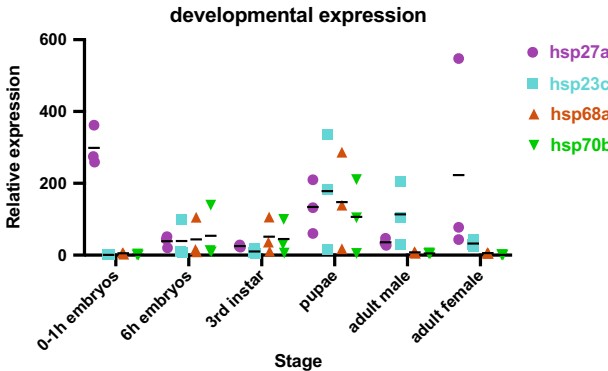

**Fig. 7 Developmental expression profiles of hsp genes.** Normalized expression values (transcript per million mapped) at different stages of development for selected small *hsp* and *hsp68/70* genes. Relative expression values for each replicate (*n* = 3) shown with mean.

annotations from the gene prediction or a blastp search of the gene prediction protein data (significance 1e-05; Supplementary Data 10). In addition, a cluster of four genes encoding cecropin-like peptides was found in one contig through manual blast searches. There were 21 matches to the 40 antimicrobial peptides, 12 of the 16 genes in cell cycle, 27 of the 32 genes in humoral response, 37 of 45 in the immune deficiency (imd) pathway, 10 of 19 in the JAK/STAT pathway, 9 of 9 in JNK pathway, and 49 of

71 in the Toll pathway. Some of the genes are components of more than one pathway. We next analyzed the developmental expression profiles of the immune response genes (Supplementary Fig. 7). In general, the expression patterns were similar to that reported for the orthologous gene in *Drosophila*[35]. For example, the antimicrobial cecropin genes are strongly expressed in pupae. It will be of interest to compare the expression profiles of immune response genes of larvae in culture (this study) with larvae taken from a wound environment.

**Genes for building sexing strains for genetic control of screwworm.** One component of a TESS is the tetracycline transactivator (tTA) driven by a gene promoter active mostly in early embryos (Supplementary Fig. 8). The second component is the proapoptotic gene *hid* driven by a tTA-regulated promoter. The *hid* gene contains the sex-specific intron from the *transformer* gene[36] and consequently only females make HID protein due to sex-specific RNA splicing (Supplementary Fig. 8). Therefore, we placed our emphasis on genes expressed strongly in 0–1 h embryos and less so at other stages, sex-specific genes or genes that positively regulate apoptosis.

**Genes expressed in the early (0–1) embryo.** The RNA collected from very early stage embryos will mostly be maternally derived but with some transcripts from zygotic genes that are activated early in embryogenesis. Cellularization occurs at ~1.5 h after egg laying. By comparison with other stages, particularly adult females, we sought to identify genes that are mostly active in the

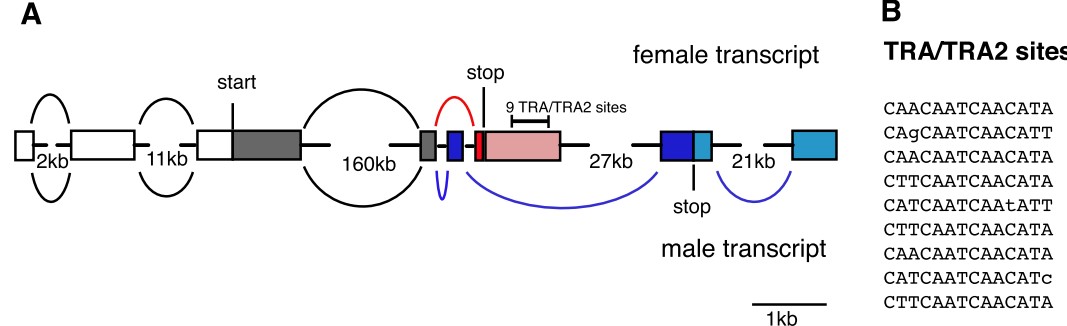

**Fig. 8 *C. hominivorax doublesex* gene organization and sex-specific transcripts. a** The *Chdsx* gene consists of four exons common to both sexes (gray or open boxes), one female-specific exon (red box) and three male-specific exons. Introns are shown with a black line with a gap indicating the intron is larger than shown. Otherwise all exons and introns are drawn to scale. Translation start and stop codons are indicated. Female and male splicing patterns are shown above and below the gene respectively. The location of the predicted TRA/TRA2 sites is indicated above the 3′UTR of the female-specific exon. **b** The sequences of the nine predicted TRA/TRA2 sites that are at least a 13 out of 14 match to the consensus sequence 5′-CAACAATCAACATA-3′ are shown.

early zygote. Seventy genes were identified that were predominantly expressed in 0–1 h embryos with little expression at other stages (Supplementary Data 5). Among the list are orthologs of the *Drosophila* zygotic cellularization genes *slow as molasses* (*slam*) and *halo*. Cellularization gene products play important roles in the formation of the cellular blastoderm[37]. We have used promoters from the cellularization genes *bottleneck* and *nullo* to make *L. cuprina* TESS[6,7]. Other genes identified include genes important for development of the early zygote such as *giant*, *zerknult* and *smoothened*. The gene list also includes all three members of the Elba complex, Bsg25A, Elba 2, and Elba3. The heterotrimeric Elba complex is required for chromatin boundary function during early embryogenesis[38]. In general, genes that are expressed highly in 0–1 screwworm embryos are also expressed highly in 0–4 h *Drosophila* embryos (Supplementary Data 5). Lastly, there are several genes that match hypothetical proteins with no known function. It will be of interest to determine if these genes are important for early development in screwworm.

In scanning the list of genes expressed in 0–1 h embryos, it is apparent that several screwworm genes are orthologous to the same *Drosophila* or *L. cuprina* gene. These include the cellularization genes, *slam* and *halo* and genes that match the hypothetical protein FF38_12096 from *L. cuprina*. Ten of the predicted *C. hominivorax* genes show similarity to *slam* and three to *halo*. The 10 *slam*-related genes are closely linked and arranged in a head-to-tail array (Supplementary Fig. 9). The first two genes in the cluster show the highest similarity to *Drosophila slam* (42% identity at the protein level). A comparison with embryo transcripts revealed a single-nucleotide error in the first gene, g13492. Correcting for this error reveals a single long open reading frame spanning g13492 and g13493 that encodes the full-length ChSLAM protein. The other eight downstream genes encode shorter proteins with little similarity to *Drosophila* SLAM (<15% identity) or to each other. The three *halo*-related genes are also closely linked in a head-to-tail array (Supplementary Fig. 9). The first gene, g6449 shows the highest similarity to *Drosophila halo* (protein is 47% identical) and is likely the *halo* ortholog. The downstream g6450 and g6451 genes show much lower similarity to *Drosophila halo* (28% identical) but the encoded proteins are 89% identical suggesting either a recent duplication or conserved function. There are 20 *C. hominivorax* genes that show a blast match to the *L. cuprina* FF38_12096 protein. The genes are in head-to-tail arrays in five contigs (Supplementary Fig. 10). Sixteen of the encoded proteins show very high identity to each other (94–100%) and high identity to FF38_12096 (~50%). The

coding region (297 bp) is a fraction of a larger sequence of ~3 kb that is repeated several times in each of the contigs. Thus, it is possible that the transcripts are not functional but simply due to transcription of one or more of the repeats.

**Sex determination genes**. In *L. cuprina*, as in *Drosophila*, Transformer (TRA) combines with Transformer2 (TRA2) to regulate the splicing of transcripts from the *doublesex* (*dsx*) and *fruitless* (*fru*) genes[39,40]. The *C. hominivorax tra* gene was reported previously[36] and the sex-specific intron was used to produce *L. cuprina* and *C. hominivorax* transgenic sexing strains[5,41]. Here we analyzed the *C. hominivorax tra2*, *dsx*, and *fru* genes. The ~4.5-kb *Chtra2* gene contains eight exons and encodes a 262 aa protein that shows 84% identity to *L. cuprina* TRA2 and 43% identity to *D. melanogaster* TRA2. The RNA-binding domain shows the highest conservation. As in *Drosophila*, the *C. hominivorax dsx* and *fru* genes are large complex genes with sex-specific alternatively spliced transcripts. The *Chdsx* gene organization is most similar to *L. cuprina*[39] and housefly *dsx* genes[42]. The first four exons are present in transcripts in both sexes. One exon is only present in female transcripts whereas three exons are male-specific (Fig. 8). There are nine predicted TRA/TRA2 binding sites within the 3′UTR of the female exon (Fig. 8). This is expected as TRA/TRA2 binding to the *dsx* precursor RNA enhances the use of the weak female-specific exon splice acceptor site[43]. The splice acceptor site for the female exon (5′-TTTTTTCTTGTGTATCACAAATTTAG|GG-3′) has several purines in the polypyrimidine tract as found in *dsx* genes in other flies. Sex-specific *dsx* transcripts were first detected in third instar larvae. Alignment of transcripts to the *Chfru* gene suggests that, as in *Drosophila*, the gene is particularly complex with multiple transcription start sites and termination sites. In *Drosophila* and housefly, sex-specific transcripts arise from the most 5′ promoter[44,45]. Similarly, analysis of the RNAseq data indicates that there are exons well upstream of the main protein-coding exons. However, the exon–intron structure for the beginning of the gene is unclear due to low coverage of RNAseq reads. One of the upstream exons identified contains four predicted TRA/TRA2 binding sites and thus could be the exon that is sex-specifically spliced. Further analysis will be required to confirm the structure and sex-specific expression of the *C. hominivorax fru* gene.

**Genes that positively regulate apoptosis**. We searched for orthologs of the 70 *D. melanogaster* genes identified by the GO

term "positive regulation of apoptotic process", GO: 0043065, identifying 55 genes (Supplementary Data 11). Included within the list were orthologs of the proapoptotic genes *hid*, *reaper* (*rpr*), and *grim*. In *D. melanogaster* and the medfly *Ceratitis capitata*, the three genes are linked along with *sickle* (*skl*)[19]. Moreover, the order of genes is conserved with the 5′–3′ order of *skl*, *rpr*, *grim*, and *hid*. In *C. hominivorax*, *skl* and *rpr* are linked in one contig while *hid* and *grim* are linked in another contig (Supplementary Data 1). *grim* and *skl* appear to be mostly expressed in 6 h embryos with few transcripts detected at other stages. *hid* and *rpr* transcripts were detected at all stages with highest levels seen in pupae. Identification of orthologs of proapoptotic genes will facilitate development of transgenic sexing strains.

**Genes important for building homing gene drive strains.** For population suppression, a Cas9-based gene drive strain would contain Cas9 driven by a germline promoter and one or more gRNAs driven by a promoter for a small RNA gene such as *U6* (Supplementary Fig. 11)[46]. The Cas9/gRNA complex would target a gene required for female fertility or fecundity. Germline promoters that have been used for gene drive strains in mosquitoes include *vasa*, *nanos* (*nos*), and *zero population growth* (*zpg*)[47–49]. *C. hominivorax* orthologs of *vasa* and *nanos* were identified in the predicted gene set. As in *Drosophila*, transcription appears to initiate well upstream of the *Chvasa* protein-coding exons (Supplementary Fig. 12). Alignment of assembled transcripts identified three potential transcription start sites. In *Drosophila* the *vasa intronic gene* (*vig*) occurs between the first exon of *vasa* and the downstream protein-coding exons[27]. The location of the *vig* ortholog between the most 5′ exons and the *Chvasa* protein-coding exons (Supplementary Fig. 12) suggests transcription begins at the more 5′ exons. As in *Drosophila*, the *Chnos* gene is very closely linked to the ortholog of the *CG11779* gene in a head to head arrangement. The *Chnos* gene promoter appears to overlap with the *ChCG11779* 5′ UTR (Supplementary Fig. 13). An ortholog of *zpg* was not identified in the predicted gene set. The closest match to the *Drosophila* ZPG protein was annotated as innexin5 (inx5). Innexins are required for gap junction function. ZPG is also known as inx4. In *Drosophila* the *zpg* gene is on chromosome 3L and very closely linked to the *nudel* gene, whereas *inx5* is on the X chromosome. The gene identified as *inx5* gene is closely linked to the *nudel* ortholog (Supplementary Fig. 12). Given the synteny observed in higher flies[50,51], the gene initially identified as *inx5* would appear to be the ortholog of *zpg*. The *Chzpg* gene is relatively simple, which could make it easier to identify a functional promoter. *Chnos*, *Chvas*, and *Chzpg* transcripts were abundant in adult females and 0–1 h embryos, which is consistent with expected expression in ovarian nurse cells and deposition in the developing oocyte. A search for *U6* snRNA genes identified five linked genes on one contig (1336) and four genes on another (9369). As the contigs show extensive similarity this is likely an assembly error due to heterozygosity in the genomic DNA rather a duplication event. The five *U6* genes are closely linked (Supplementary Fig. 14).

Suppression of cage populations of the malaria vector *Anopheles gambiae* was achieved by using a homing gene drive targeting the highly conserved female-specific exon of the *dsx* gene[49]. Thus, the *Chdsx* gene described above would be an excellent target for a gene drive. Other genes targeted in mosquito cage experiments were the orthologs of the *Drosophila* female fertility genes *yellow-g* and *nudel*[52]. The ortholog of *nudel* is closely linked to *zpg* (see above) and the *yellow-g* ortholog is a single-copy gene (g18385) on a different scaffold.

## Discussion

The whole-genome assembly of the New World screwworm reported in this study will serve as a reference for future genetic investigations of this obligate parasite of warm-blooded animals. For example, studies on screwworm population structure in countries where screwworm remains endemic, will facilitate identification of the point of origin of flies in any outbreaks, such as occurred in Florida in 2016[53]. BUSCO analysis suggests that the genome is of high quality with few genes absent from the assembly and little fragmentation or duplication. Nevertheless, the assembled size is 94 Mb larger than measured, which could indicate assembly errors due to residual heterozygosity in the highly inbred strain that was the source of genomic DNA. We identified orthologs of genes that are known to play important roles in transcription, sex determination, apoptosis, chemosensation, heat shock, and immune response. We also identified genes that appear to be unique to *C. hominivorax*. These genes are mostly active in the embryo and adult males. It will be of interest to determine if the latter are important for male reproduction or mating performance, since male mating performance is critical for any genetic control program[5]. Functional analysis of the male-specific genes and other genes of interest such as *Chdsx* and *Chfru* will benefit from our recent work developing efficient CRISPR/Cas9 gene editing technologies in screwworm[54].

Along with annotation and analysis of the genome, we conducted, to our knowledge, the first developmental RNA-seq analyses for *C. hominivorax*. In precellular embryos, RNA-seq studies identified genes that are activated in the early zygote. These include genes that are known to play important roles in cellularization and patterning. We found additional copies of two of the cellularization genes, *slam* and *halo*, that were expressed in early embryos. The eight *slam*-related genes encoded smaller proteins with low similarity to the *Drosophila* SLAM protein. Of the three genes that were similar to *Drosophila halo*, one appears to be the true ortholog and the other two encode more distantly related proteins. Functional analyses will be needed to determine if these genes are required for cellularization.

To study the molecular evolution of protein-coding genes in *C. hominivorax*, we estimated the ratio of non-synonymous to synonymous substitutions (d$N$/d$S$) of genes with orthologs in 11 dipteran species. The d$N$/d$S$ ratio test revealed a strong purifying selection. Although the d$N$/d$S$ ratio is regarded as a conservative test for positive selection, our analysis indicated 40 genes with an $\omega > 1$, providing evidence of positive selection in *C. hominivorax*. Almost half of these genes had little or no annotated function. Non-annotated genes have been previously described to be less constrained and with lower *P*-values for the d$N$/d$S$ ratio tests compared to genes with annotated functions[26]. It has been proposed that these genes with no assigned annotations may have an important, yet undiscovered, role in evolution[26]. The orthologs of *Dh31-R*, *S6k*, *robo1*, and *crim* are of particular interest for future functional studies given their potential roles in larval temperature regulation, larval feeding, male behavior, and tracheal development respectively. Additionally, several of the unclassified genes have significant expression in male reproductive organs for *C. hominivorax* and *Drosophila*, suggesting critical roles in male fertility.

Cost is a significant barrier for implementation of SIT programs to suppress or eradicate *C. hominivorax* populations in the Caribbean or South America. The genes identified in this study could be used to build TESS, which would have reduced rearing costs since females die early in development. Releases of fertile males carrying dominant female lethal genes should be more efficient than SIT. However, very efficient strains would be needed for control in South America. Importantly, we identified genes that could serve as the basis for developing homing gene

drive strains that target genes required for female development (*Chdsx*) or female fertility (*Chnudel*).

## Methods

**New World Screwworm rearing**. The J06 wild-type strain of *C. hominivorax* was reared at the COPEG biosecurity plant in Panama as described previously[5]. To obtain the highly inbred line, 20 crosses were performed between single males and single virgin females. In all, 16 of these initial crosses were fertile. From the off-spring of each cross a single male and single virgin female were randomly selected and crossed. The process was repeated for 10 generations so that the final highly inbred line strain of *C. hominivorax* was obtained after 10 generations of single pair mating. From the highly inbred line, several stages of development were collected and rapidly frozen in liquid nitrogen. Three independent samples were collected for each stage. The stages collected were 0–1 h and 6 h embryos, 72 h wandering 3rd instars, 1-day-old pupae and 6-day-old adult male and female.

**Nucleic acid isolation and sequencing**. High molecular weight DNA was isolated from mixed sex 6 h embryos of the inbred strain using procedures described previously[55] but taking additional care to minimize DNA shearing. The frozen embryos were ground to powder with a mortar and pestle under liquid nitrogen and then suspended in 4 mL STE buffer (50 mM Tris-HCl, pH 7.5, 100 mM NaCl, 10 mM EDTA, and pH 8). In all, 200 μL 10% SDS and 8 μL RNase A (Cat# R4642 Sigma Aldrich St. Louis, Missouri) were added and samples were incubated at 56 °C. After 30 min, Proteinase K (Cat# P2308 Sigma Aldrich) was added to 100 μg/mL and the sample was incubated overnight at 56 °C. In total, 3 mL phenol:chloroform: isoamyl alcohol [25:24:1] (Cat#P2069, Sigma) was added and samples were rotated gently for 10 min at room temperature (RT). Samples were then centrifuged 10 min at 3000 RPM at 4 °C. The aqueous layer was transferred to a new tube. The extraction was repeated. One tenth volume 3 M sodium acetate, pH 5.2, and 2 volumes cold 100% ethanol were added and the samples were inverted 2–3 times. The samples were incubated at −20 °C overnight and then centrifuged for 30 min at 6000 RPM at 4 °C. The supernatant was removed from the pellet and 1 mL cold 75% ethanol was added. The samples were centrifuged 10 min at 6000 RPM at 4 °C. The supernatant was removed from the pellet and it was allowed to air dry 10 min. The last of the supernatant was removed and the pellet was allowed to air dry 10 additional minutes before being resuspended in 50–100 μL TE Buffer. The average size of the DNA was estimated by using agarose gel electrophoresis with a pippin pulse power supply (Sage Science) following conditions recommended by the manufacturer. DNA that was greater >200 kb in size was used for preparation of 20 and 30 kb DNA libraries following instructions from the manufacturer (Pacific Biosciences). Five SMRT cells were run from the 30-kb library which produced 2592 Mb of sequence with an average read length of 9281 bp. Twenty SMRT cells were run from one the 20-kb libraries at the Yale Center for Genome Analysis which produced 14176 Mb of sequence with an average read length of 8767 bp. Seventeen SMRT cells were run from one of the 20-kb libraries at RTLGenomics which produced 22487 Mb of sequence with an average read length of 10044 bp. We also obtained single-end 125 bp reads with the Illumina HiSeq 2500 on adult male and female DNA resulting in ~225 million reads for the female sample and 203 million reads for the male sample. Trimming of adapter and for quality was done using Trimmomatic version 0.23[56] with a sliding window quality cut-off of 15 and a minimum length of 36 required to keep a read. Trimming removed less than 1% of each dataset, resulting in a total of ~224 million reads for the Female sample and ~202 million reads for the male sample.

Total RNA was extracted from frozen samples using a Qiagen RNeasy mini kit (Qiagen, USA) using the manufacturers recommended procedures. RNA integrity, purity, and concentration were assessed using an Agilent 2100 Bioanalyzer with an RNA 6000 Nano Chip (Agilent Technologies, USA). Purification of messenger RNA (mRNA) was performed using the oligo-dT beads provided in the NEBNExt Poly(A) mRNA Magnetic Isolation Module (New England Biolabs, USA). Complementary DNA (cDNA) libraries for Illumina sequencing were constructed using the NEBNext Ultra Directional RNA Library Prep Kit (NEB) and NEBNext Mulitplex Oligos for Illumina (NEB) using the manufacturer-specified protocol. The amplified library fragments were purified and checked for quality and final concentration using an Agilent 2200 Tapestation with a High Sensitivity DNA chip (Agilent Technologies, USA) and a Qubit fluorometer (ThermoFisher, USA). The final quantified libraries were pooled in equimolar amounts for clustering and sequencing on an Illumina HiSeq 2500 DNA sequencer, utilizing a 125-bp single end sequencing reagent kit (Illumina, USA). A total of 451,814,971 single-end 125 bp reads were produced on the HiSeq across the different life stages. An additional 249 million 150-bp paired-end reads and 19.6 million 300 paired-ends reads for the early Embryo stage were run on the NextSeq and MiSeq, respectively. Trimming of adapter and for quality was done using Trimmomatic version 0.23[56] with a sliding window quality cut-off of 15 and a minimum length of 36 required to keep a read. Trimming removed <1% of each dataset, resulting in a total of 716,463,678 reads for use across all samples.

**Genome assembly and gene prediction**. Canu version v1.4 (+62 commits) r8057 (fb27dc42fa749cc6c4245bc6ec7162f2c7760612) was used to assemble the PacBio data[57]. Default parameters produced an assembly size much larger than was expected (842 Mbp), likely due to the presence of heterozygous alleles that were assembled as separate contigs. Canu parameters were modified to be more permissive of allelic variation using the following options: '-genomeSize = 440 m -obtovlErrorRate = 0.15 -obtErrorRate = 0.15 -obtovlerrorrate = 0.15 -errorRate = 0.05 -pacbio-corrected'. Reassembly with the modified parameters merged some alternative alleles and resulted in a reduced assembly size of 532 Mbp. This assembly was then polished using Arrow version 2.1.0 resulting in a final assembly size of 534 Mbp with a maximum contig size of 5.46 Mbp and a contig NG50 of 0.77 Mbp (assuming a haploid genome size of 440 Mbp).

RNA reads were mapped to the reference genome with Tophat version 2.1.1[58]. The mapped reads were used as input for Braker v1.9[59], which calls GeneMark-ET (v4.33)[60] to create a training set for Augustus (v3.2.3)[61,62]. The resulting initial gene prediction set contains 22,491 protein-coding genes, of which 20,975 have a match to a protein in the NCBI NR database.

**Repetitive DNA sequences**. To identify interspersed repetitive sequences, RepeatModeler2[63] was used to construct a de novo repeat library for *C. hominivorax*. RepeatMasker v. 4.0.7[64] was then run using the de novo *C. hominivorax* repeat library that was combined with Drosophila repetitive sequences extracted from the RepeatMasker library and a combination of the DFam_Consensus and RepBase databases.

**Comparative genomics**. A species phylogeny based on 612 protein sequences (231,305 amino acids) was reconstructed to determine the evolutionary relationships among nine dipteran species. The official protein set of *Lucilia cuprina* (NCBI, GCF_000699065.1), *Musca domestica* (NCBI, GCF_000371365.1), *Sarcophaga bullata* (PRJNA476317), *Stomoxys calcitrans* (PRJNA288986), *Glossina morsitans* (VectorBase, GmorY1.8), *Drosophila melanogaster* (NCBI, GCF_000001215.4), *Mayetiola destructor* (i5k, Mdes_1.0), *Aedes aegypti* (Vector-Base, AaegL3.3), and *Anopheles gambiae* (VectorBase, AgamP3) were downloaded from NCBI, VectorBase, or i5k, and searched against the *C. hominivorax* gene set using BLASTp. A significant e-value cut-off ≤1e$^{-5}$ was applied and only genes that had a single hit across all eight species were included in further analysis. A total of 612 individual proteins were aligned with MAFFT[65] using default settings, and alignments were trimmed using gBlocks to remove gaps[66]. The aligned single-copy protein-coding genes were then concatenated and the phylogeny was reconstructed using RAxML version 8.2.8[67,68] with the PROTGAMMAWAG model and 100 bootstrap replicates. The phylogeny was visualized with FigTree version 1.4.2 (http://tree.bio.ed.ac.uk/software/figtree/). Orthologous groups of genes were also determined among the nine species using OrthoFinder (v 2.2.7)[69] using default settings. Transcription factors were identified based on methods used for other invertebrate genomes[21]. In brief, putative TFs were identified by scanning the amino acid sequences of all proteins for putative DNA-binding domains using the HMMER software package[70] and a compilation of Pfam DNA binding domain models[71]. Expression profiles of each TF to determine specific TF candidates are associated with sex and development.

For the molecular evolution analysis of protein-coding genes, we used a slightly modified pipeline to search for orthologs[72]. We used the official protein set of the nine species previously employed for the phylogenetic reconstruction (*L. cuprina*, *M. domestica*, *S. bullata*, *S. calcitrans*, *G. morsitans*, *D. melanogaster*, *M. destructor*, *A. aegypti*, and *A. gambiae*), and included two other taxa, *Ceratitis capitata* (NCBI, GCA_000347755.2) and *Phormia regina* (NCBI, SAMN05567884) to search for one-to-one orthologs among them and *C. hominivorax*. After all-by-all blast, the results were filtered by hit fraction length (0.4). Sequences recovered by blast were then clustered with MCL[73] with an inflation value of 1.4, and all sequences in a cluster are aligned with MAFFT[74]. Maximum-likelihood phylogenies were inferred for each cluster using RAxML v7.3.5[67] and only genes without any taxon repeat in their gene tree were kept for further analyses[72]. For all genes we used the coding sequences (CDS) with a minimal taxon occupancy of five, i.e. only single-copy orthologs found in at least five of the analyzed species (*C. hominivorax*, and any other four).

Inferred orthologs were used for the analyses of molecular evolution of *C. hominivorax* coding sequences. All alignments and alignment cleaning were done with TranslatorX[75]. CDS were translated into amino acids and aligned with MUSCLE[76]. Aligned amino acid sequences were back-translated, and poorly aligned regions were masked with GBlocks also implemented in TranslatorX. These alignments were used as inputs to estimate rates of synonymous ($d_S$) and non-synonymous substitution ($d_N$), and their ratio ($\omega$) with codeml in PAML 4.7[77].

Three models were tested: (a) a neutral model with a fixed $\omega = 1$ (null hypothesis), (b) a free model with an $\omega$ for each branch of the tree (model 1), and (c) a two-ratios model with an $\omega$ for the *C. hominivorax* branch and an $\omega$ calculated for the background. Model 1 and model 2 were compared to the null hypothesis with the likelihood ratio test (LRT). Alignments producing $d_S = 0$, and/or $d_N/d_S > 10$ were discarded. LRT was done with a $\chi^2$ test with $\alpha$ set at 0.05, and the false discovery rate (FDR) was used for multiple test correction. The LRT test indicates which model fits better our $d_N/d_S$ estimations. Thus $\alpha$ determines which genes are evolving under neutrality (null hypothesis) or a selective regime. If neutrality is rejected, then $d_N/d_S$ values indicate purifying selection ($d_N/d_S < 1$) or positive selection ($d_N/d_S > 1$). We compared the distribution of the dN/dS ratios across categories of genes by using GO terms. Only GO terms with at least 30 genes

were used for this comparison, resulting in 30 terms for the "Biological Process" category, 15 terms for the "Cellular Component" category, and 32 terms for the "Molecular Function" category.

**Chemosensory gene analyses.** Odorant, gustatory and ionotropic receptors (ORs, GRs, and IRs), and odorant-binding proteins (OBPs) were manually annotated by BLASTn and tBLASTn analysis of the genome and transcriptome (*C. hominivorax*) assemblies using *D. melanogaster*[27] and *M. domestica*[78] gene models. BLAST analysis of the *C. hominivorax* assemblies was conducted using Geneious v6.1.8 (https://www.geneious.com). Chemosensory genome annotation was conducted following protocols described by Hickner et al.[79].

Gene models were evaluated with the aid of a multiple alignment and phylogenetic tree for each of the four gene families in *D. melanogaster*, *M. domestica*, *S. calcitrans*[80], and *C. hominivorax*. Peptide sequences were multiply aligned then visualized using ClustalX 2.1[81,82]. The maximum likelihood method in RAxML with the PROTGAMMAAUTO model option and 500 bootstrap replications was used for phylogenetic analysis[68]. Trees were visualized and figures generated using the interactive Tree of Life (iTOL) v4 software[83].

Several single nucleotide insertions/deletions (indels) were found in the *C. hominivorax* genome assembly that were not present in the corresponding transcripts. These were considered functional genes and indicated by "_fixT" in Supplementary Data 4. Indels that could not be evaluated with the corresponding transcript were considered pseudogenes and suffixed with "P". Due to the complex and sometimes ambiguous orthologous/paralogous relationships among the members of these gene families, we named them based on their position on the scaffolds/contigs. The exceptions include the odorant receptor co-receptor (*Orco*) and the putative $CO_2$ and sugar receptors where we followed the convention[78]. In addition, several IRs with clear one-to-one orthologous relationships in the species used for comparative analysis were named in accordance with the *D. melanogaster* orthologs, and the rest were named as per the contig/position[78].

**Identification of transcription factors.** We identified likely transcription factors (TFs) by scanning the amino acid sequences of predicted protein-coding genes for putative DNA binding domains (DBDs), and when possible, we predicted the DNA binding specificity of each TF using the procedures described in Weirauch et al.[84]. Briefly, we scanned all protein sequences for putative DBDs using the 81 Pfam[85] models listed in Weirauch and Hughes[71] and the HMMER tool[70], with the recommended detection thresholds of Per-sequence Eval < 0.01 and Per-domain conditional Eval < 0.01. Each protein was classified into a family based on its DBDs and their order in the protein sequence (e.g., bZIPx1, AP2x2, Homeodomain+Pou). We then aligned the resulting DBD sequences within each family using clustalOmega[86], with default settings. For protein pairs with multiple DBDs, each DBD was aligned separately. From these alignments, we calculated the sequence identity of all DBD sequence pairs (i.e. the percent of amino acid residues that are exactly the same across all positions in the alignment). Using previously established sequence identify thresholds for each family[84], we mapped the predicted DNA binding specificities by simple transfer. For example, the DBD of g19927.t1 is 87.7% identical to the *Drosophila melanogaster* Antp protein. Since the DNA-binding specificity of Antp has already been experimentally determined, and the cutoff for the homedomain family of TFs is 70%, we can infer that g19927.t1 will have the same binding specificity as Antp.

Using the above procedure, we identified a total of 982 putative TFs in the *C. hominivorax* genome, representing 4.7% of the total number of *C. hominivorax* genes. This fraction is similar to that seen in related species, such as *Drosophila melanogaster* (5.5%), *Bombyx mori* (5.3%), and *Danaus plexippus* (4.9%). The distribution of *C. hominivorax* TFs across families is similar to that of other insects (Fig. 3). Of the 982 *C. hominivorax* TFs, we were able to infer motifs for 389 (40%) (Supplementary Data 7), mostly based on DNA binding specificity data from *D. melanogaster* (355 TFs), but also from species as distant as human (23 TFs). Many of the largest TF families have inferred motifs for a substantial proportion of their TFs, including Homeodomain (104 of 106, 98%), bHLH (67 of 67, 100%), and nuclear receptors (27 of 29, 93%). As expected, the largest gap is for $C_2H_2$ zinc fingers (only 43 of 351, ~12%), which evolve quickly by shuffling their many zinc finger arrays, resulting in largely dissimilar DBD sequences across organisms[87].

**Developmental expression analyses.** RNA-seq analyses were conducted according to Attardo et al.[21]. Briefly, Illumina reads were examined for quality with FASTQC and trimmed/cleaned with Trimmomatic to remove low quality reads. Reads were mapped to the predicted genes with the use of CLC Genomics (Qiagen), allowed two mismatches with over 90% similarity. Expression values were converted to transcript per million mapped (TPM). EdgeR[88] was utilized for statistical comparison under standard setting with a false detection rate (FDR) at 0.001. Genes with over twofold enrichment or reduction were used to create development pairwise comparisons. Genes that were enriched in a single developmental stage compared to all other stages were deemed stage-specific. Heat maps for stage-specific genes sets were produced with the "pHeatmap function" in R. Putative function was assigned through BLAST comparison to proteomes of other flies. Lastly, GO analyses were conducted by examining the *Drosophila melanogaster* orthologs with g:Profiler[89] and visualization of the GO results through REVIGO[90].

Along with our comparative analysis between each set, a gene co-expression network was created utilizing the WGCNA[91] R software package (https://horvath.genetics.ucla.edu/html/CoexpressionNetwork/Rpackages/WGCNA/). This approach was used to correlate genes that are similarly expressed across developmental stages and sexes, place them into modules, and relate the resulting modules to the samples.

To prepare our expression data for WGCNA, genes of zero variance were removed (e.g. expression values of 0 for all sets) from the dataset, leaving 11217 genes for an unsigned network construction. A soft thresholding power of 10 was chosen based on the scale-free topology fit index curve made prior to network construction. The minimum module size for this network was set to 20. To indicate identity of each module to the developmental stages, the developmental stages were used as input traits during network construction. The modules exhibiting the greatest significance to the trait data (<0.05) were further analyzed to determine function and relationship to the developmental stages and sexes GO analyses were conducted as before by examining the *Drosophila melanogaster* orthologs with g:Profiler[89].

**Statistics and reproducibility.** For the developmental gene expression analysis, three independent samples were collected for each stage and all datasets have been made public on the NCBI Sequence Read Archive. CLC Genomics was used for mapping of reads to predicted genes. Statistical analyses and generation of figures for RNA-seq studies were performed in the R software environment and described in each section of the paper. The package 'dplyr' was used for large data manipulation and the package 'ggplot2' was used for creating graphics. Mann–Whitney tests were done with the 'wilcox.test' function of the core R package. The significance level, alpha, was set to 0.05, and false discovery rate correction was done for multiple tests. Other specific tests used in this study are described in the relevant section (e.g. comparative genomics).

**Reporting summary.** Further information on research design is available in the Nature Research Reporting Summary linked to this article.

## Data availability
The genbank accession for the genome assembly is ASM430292v1. The Illumina and PacBio reads from genomic DNA and the RNA sequencing data were deposited at NCBI under SRA accession PRJNA641284. The datasets supporting the conclusions of this article are included within the article and its supplementary files. The supplementary data files are available at Dryad, https://doi.org/10.5061/dryad.x69p8czg2.

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

## Acknowledgements

We are grateful for technical assistance from Nicolas Mendoza, Domitildo Martinez, Rosaura Sanchez, Hermogenes Gonzalez, Brigido Gonzalez, and Jaime Ceballos at the ARS-COPEG laboratory. We thank Ying Yan for the transgenic embryonic sexing strain figure (Supplementary Fig. 8). Funding is gratefully acknowledged from specific cooperative agreements between the USDA-ARS and NCSU and from the Panama-United States Commission for the Eradication and Prevention of Screwworm (COPEG) and USDA-APHIS. Mention of a proprietary product does not constitute endorsement or recommendation for its use by the USDA. USDA is an equal opportunity provider and employer.

## Author contributions

M.J.S. designed research, performed research, analyzed data, and obtained funding for this project. R.J.D. isolated high molecular weight DNA and organized Illumina DNA sequencing. Differential RNA expression analyses were performed by S.B., V.V., E.O.M., and J.B.B.; J.B.B., G.A.C., and T.T.T. performed comparative genomics analyses. M.T.W. performed transcription factor analyses. Assembly of the genome from PacBio reads was performed by A.M.P. Gene predictions and BUSCO analyses were performed by E.H.S. Chemosensory gene analyses were conducted by P.V.H. and Z.S.; A.S., G.Q., and M.V. performed crosses to create the inbred strain and then collected samples for nucleic acid isolation. S.R.S. supervised research in Panama and obtained funding for this project. M.J.S., P.V.H., T.T.T., E.H.S., and J.B.B. wrote and edited the manuscript. All authors read and approved the final manuscript.

## Competing interests

The authors declare no competing interests.
