## [Peer Review File · Communications Biology]

Reviewers' comments:

Reviewer #1 (Remarks to the Author):

Thank you for your valuable and interesting study.

Some Calliphoridae species do cause opportunistic myiasis to dead tissues, but *Cochliomyia hominivorax* is one of unusual attackers to live healthy animals.

I think the background, methods, results, and conclusion were all well written.

This research can be an important reference for developing strategies to control the livestock pest caused by *H. hominivorax*.

In particular, a gene for tracheal development, *crim* is one of the genes under positive selection, because maggots of this fly species have distinctively thick and black trachea visible from the surface and this feature is one of the diagnostic features to distinguish the larvae of this species from the larvae of *C. macellaria*, the secondary screw worm which does not show thick and black trachea.

Because thick and black tracheas are frequent finding among parasitic fly larvae, evolutionary significance of this gene among different parasitic fly species may give an interesting result.

Additionally, parasitic fly larvae tend to have well developed spines or rough outer surfaces maybe related to their needs to fix in live animal tissues.

Therefore, I am very curious whether any gene under positive selection is related to spine formation of larvae or surface consistency.

Reviewer #2 (Remarks to the Author):

In this study, the authors assembled the whole genome of the screwworm fly and annotated the genome. They also obtained transcriptomic data across developmental stages. They conducted comparison of genomes among fly species, analysis of chemosensory genes and transcription factors, and analysis of gene co-expression networks with WGCNA (Weighted Gene Co-expression Network Analysis). This study produced a large amount of data for the genes and gene expression in the screwworm fly potentially useful for development of control tactics of the important pest species. Nonetheless, the present manuscript is largely descriptive and difficult to follow because the description but is not well organized for understanding by readers who are not familiar with highly specific analyses used in the present study. The figures require substantial improvement to make readers easily understand the contents.

Tables 1 and 2 are not necessary because all contents can be written in the text.

Figure 1 with circular tree is hard to see. Use a ladderized tree with taxon name and information on the right.

Similarly, the contents of figure 2 are hardly understood. The graphs showing the number of enriched genes for every comparison is hard to see and useless, and the graphs indicating enriched GO terms may be replaced with a table, which will be easier to understand.

In Figure 3 B (and figures 5 and 6), gene IDs are indicated, but without any explanation. We cannot see what are the grouping of genes, though it seems that the genes were grouped by the similarity of expression pattern.

Figure 4 is very hard to see, and the legend is confusing; Legend for B says "Relationship between each module", but is this "between all genes"? This panel lacks legend for the heat map. In C, the heat map seems to represent correlation coefficient, but there is no explanation and the legend of heat map cannot be read. There is no exact explanation for GO terms listed and "frequencies" indicated.

L297-299: 4489 - 14 = 4475; perhaps 4476 in L299 should read 4475.

L552: remove "between the" (duplicated)

Reviewer #3 (Remarks to the Author):

This paper describes an assembly of the genome of the New World Screwworm fly (*Cochliomyia hominivorax*) as well as a characterization of gene expression during several developmental stages and an analysis of the selection pressures that have acted on the fly genes during the evolution of Diptera. The assembly appears to be of relatively good quality, the paper globally well written and presented. Overall, I believe the data presented in this manuscript will be an interesting resource to researchers involved in finding strategies to control of this livestock pest and the genome will be valuable to many researchers in the field of genome evolution. The authors will find below a number of comments or questions that should be addressed.

1 – Most if not all eukaryote genomes are replete with transposable elements (TEs) and other repeated sequences. There is no mention of this major genomic component in the paper. I personally believe that all genome papers should include at least a short section on repeated sequences, with global TE content and TE content broken down by TE superfamilies. In addition to be interesting in itself, characterizing transposable element sequences would ensure that none of the annotated genes are TEs and that none of the genes included in the analysis of selection are TEs. I strongly encourage the authors to perform de novo characterization and annotation of repeated sequences. This can at least be done using automated procedures using RepeatModeler and RepeatMasker for example (as in <https://www.g3journal.org/content/9/5/1313> for example).

2 – I am not myself an expert in dN/dS ratio analysis. However, I suspect this analysis is problematic here. In addition to the fact that high dN/dS ratio often result from various artefacts that need to be thoroughly checked (e.g. doi: 10.1038/s41559-018-0584-5), the values in additional table 16 do not seem to make sense at all. For example, the background dS for g16465 is extremely high (25.95 mutation/synonymous site) while it is 0 for Chom suggesting that there is a problem in the underlying alignment. Furthermore, while some Chom genes have dN and dS values equal to zero, their dN/dS ratio have been calculated and are high (e.g. g9400). Furthermore, the p-values given by PAML do not allow to conclude on whether a particular gene is submitted to positive selection or not. They can just be used to tell whether a model is statistically better than another.

Before even thinking of whether the biological interpretations made in the paper about the genes putatively evolving under positive selection make sense or not, I would strongly recommend to carefully check all alignments resulting in high dN/dS ratio, as well as revising additional table 16 and using other methods that are more appropriate to detect episodic positive selection acting on a given gene or branch (e.g. <https://www.datamonkey.org/>).

3 – I do not understand why the authors chose to illustrate the probability (on the x axis of A) that a model of selection is better than another for each gene. This does not give a sense of whether a gene is under strong positive selection or not (please see my comment above). This figure is also extremely hard to read and interpret.

4 – Figure 7. It is unclear to me what the Y axis of the graph illustrates (expression relative to what value ?)

5 - I could not see the right part of Figure 1 on the pdf version of the manuscript and wasn't able to identify the individual file in which this figure might have been additionally submitted. In addition, in the legend of this figure, there might be a problem with the following sentence : « Benchmarking

Universal Single-Copy Orthologs [17] analyses based on the dipteran dataset (odb8). »

6 – Is the phylogenetic position of *C. hominivorax* obtained in this study expected compared to previous analyses ?

7 – Line 586-588 : « Indeed, clusters of U6 snRNA genes were found in two contigs that showed extensive similarity and thus are likely from different alleles rather than a duplication. »

I here list a number of typos that I have spotted but it is very likely that there are many more as I have not read the entire manuscript with an homogeneous level of effort.

Line 160 : please correct « suggests »

Line 195 : replace « in number in » by « in number to » ?

Line 130 : Please correct « Our goal of these analyses »

Line 141 : Please correct « aminoglygan »

Line 242 : the repetition of « associated » does not look great

Line 252 : perhaps replace « the highest expression in levels embryos and male » by « the highest expression in embryos and males » ?

Figure 4 is overall very hard to read

Legend of Fig 5 : replace « categorizes » with « categories » ?

Line 333 : « Developmental RNA-seq two gene ontology categories under positive selection based on additional table 16. »

There seems to be a problem with this sentence.

Line 345 : replace « following » by « followed » ?

Line 385 : « males » instead of « male »

Line 390 : please correct the sentence

Line 448 : please correct « propapototic »

Line 584 : Do you mean 94 Mb ?

Response to Reviewers' comments:

Our responses are in *italics* and follow each point made by a reviewer.

Reviewer #1 (Remarks to the Author):

Thank you for your valuable and interesting study.

Some Calliphoridae species do cause opportunistic myiasis to dead tissues, but *Cochliomyia hominivorax* is one of unusual attackers to live healthy animals.

I think the background, methods, results, and conclusion were all well written.

This research can be an important reference for developing strategies to control the livestock pest caused by *C. hominivorax*.

1. In particular, a gene for tracheal development, *crim* is one of the genes under positive selection, because maggots of this fly species have distinctively thick and black trachea visible from the surface and this feature is one of the diagnostic features to distinguish the larvae of this species from the larvae of *C. macellaria*, the secondary screw worm which does not show thick and black trachea.

Because thick and black tracheas are frequent finding among parasitic fly larvae, evolutionary significance of this gene among different parasitic fly species may give an interesting result. Additionally, parasitic fly larvae tend to have well developed spines or rough outer surfaces maybe related to their needs to fix in live animal tissues.

Therefore, I am very curious whether any gene under positive selection is related to spine formation of larvae or surface consistency.

*We thank the reviewer for bringing this to our attention. The referee is correct that *crim* is one of four *Lys-6* like protein required for septate junction formation in the larval trachea. We have added this to the section of the paper that discusses the genes under positive selection in *C. hominivorax*.*

Reviewer #2 (Remarks to the Author):

In this study, the authors assembled the whole genome of the screwworm fly and annotated the genome. They also obtained transcriptomic data across developmental stages. They conducted comparison of genomes among fly species, analysis of chemosensory genes and transcription factors, and analysis of gene co-expression networks with WGCNA (Weighted Gene Co-expression Network Analysis). This study produced a large amount of data for the genes and gene expression in the screwworm fly potentially useful for development of control tactics of the important pest species. Nonetheless, the present manuscript is largely descriptive and difficult to follow because the description but is not well organized for understanding by readers who are not familiar with highly specific analyses used in the present study. The figures require substantial improvement to make readers easily understand the contents.

1. Tables 1 and 2 are not necessary because all contents can be written in the text.

In Tables 1 and 2 report the genome assembly statistics and busco scores. These are simple tables that can be easily read by someone scanning the paper. We would prefer to include

the table1 in the paper as these are the assembly stats but have incorporated table2 into the main body of the paper as suggested.

2. Figure 1 with circular tree is hard to see. Use a ladderized tree with taxon name and information on the right.

We have changed the figure as suggested to improve clarity.

3. Similarly, the contents of figure 2 are hardly understood. The graphs showing the number of enriched genes for every comparison is hard to see and useless, and the graphs indicating enriched GO terms may be replaced with a table, which will be easier to understand.

We have changed the format to highlight the main comparisons and GO terms have been added as a list.

4. In Figure 3 B (and figures 5 and 6), gene IDs are indicated, but without any explanation. We cannot see what are the grouping of genes, though it seems that the genes were grouped by the similarity of expression pattern.

We have added a note to the figure legend to indicate groups of genes are based on similarity in expression.

5. Figure 4 is very hard to see, and the legend is confusing; Legend for B says “Relationship between each module”, but is this “between all genes”? This panel lacks legend for the heat map. In C, the heat map seems to represent correlation coefficient, but there is no explanation and the legend of heat map cannot be read. There is no exact explanation for GO terms listed and “frequencies” indicated.

This figure is likely difficult to see in the review due to it being a lower quality as a consequence of inserting into the word doc for the initial submission. We have added a legend to the heat map and provided more details within the figure legend. Frequencies is now defined in the figure legend. We believe the higher quality of this figure and corrections improve the quality substantially

6. L297-299: $4489 - 14 = 4475$; perhaps 4476 in L299 should read 4475.

Corrected as suggested

7. L552: remove “between the” (duplicated)

Corrected as suggested

Reviewer #3 (Remarks to the Author):

This paper describes an assembly of the genome of the New World Screwworm fly (*Cochliomyia hominivorax*) as well as a characterization of gene expression during several developmental stages and an analysis of the selection pressures that have acted on the fly genes during the evolution of Diptera. The assembly appears to be of relatively good quality, the paper globally well written and presented. Overall, I believe the data presented in this manuscript will be an interesting resource to researchers involved in finding strategies to control of this livestock pest and the genome will be valuable to many researchers in the field of genome evolution. The authors will find below a number of comments or questions that should be addressed.

1. Most if not all eukaryote genomes are replete with transposable elements (TEs) and other repeated sequences. There is no mention of this major genomic component in the paper. I personally believe that all genome papers should include at least a short section on repeated sequences, with global TE content and TE content broken down by TE superfamilies. In addition to be interesting in itself, characterizing transposable element sequences would ensure that none of the annotated genes are TEs and that none of the genes included in the analysis of selection are TEs. I strongly encourage the authors to perform de novo characterization and annotation of repeated sequences. This can at least be done using automated procedures using RepeatModeler and RepeatMasker for example (as in <https://www.g3journal.org/content/9/5/1313> for example).

We created a de novo repeat library for the screwworm genome using RepeatModeler2. We then RepeatMasker with the de novo library. That analysis found that 45.1% of the genome was repetitive. We then manually added Drosophila repeats from RepeatMasker to the de novo screwworm library and ran RepeatMasker again. Adding the Drosophila repeats found a few additional repeats in the screwworm genome (e.g. matches to P and hAT-hobo), so we have included the summary of this analysis in the paper (Supp File 3). The total of the genome that was repetitive only increased slightly (to 45.2%) after adding the Drosophila repeats. We searched the annotated genes for TEs. We can find TEs in introns but none of the annotated coding regions appear to be TEs. We do find transcripts in the embryo and total transcriptomes that match TEs (e.g. Jockey, gypsy). But the transcripts are short and don't contain long open reading frames. So, the significance of these transcripts is questionable. We have included a paragraph on the repetitive sequences found in the screwworm genome at the end of the first section of Results on genome assembly.

Throughout this paper we have sought to connect our results to the biology of screwworm, given Communications Biology is a general biology journal. We have no doubt that repetitive sequences could be having some impact on gene expression or played a role in screwworm evolution. The importance of TEs in screwworm biology and evolution should become clearer as more blow fly genomes are assembled and annotated.

2. I am not myself an expert in dN/dS ratio analysis. However, I suspect this analysis is problematic here. In addition to the fact that high dN/dS ratio often result from various artefacts that need to be thoroughly checked (e.g. doi: 10.1038/s41559-018-0584-5), the values in additional table 16 do not seem to make sense at all. For example, the background dS for g16465 is extremely high (25.95 mutation/synonymous site) while it is 0 for Chom suggesting that there is a problem in the underlying alignment. Furthermore, while some Chom genes have dN and dS values equal to zero, their dN/dS ratio have been calculated and are high (e.g. g9400). Furthermore, the p-values given by PAML do not allow to conclude on whether a particular gene is submitted to positive selection or not. They can just be used to tell whether a model is statistically better than another.

Before even thinking of whether the biological interpretations made in the paper about the genes putatively evolving under positive selection make sense or not, I would strongly recommend to carefully check all alignments resulting in high dN/dS ratio, as well as revising additional table 16 and using other methods that are more appropriate to detect episodic positive selection acting on a given gene or branch (e.g. <https://www.datamonkey.org/>).

We understand the concern of the reviewer about the evolutionary analysis. It is true that a dN/dS analysis demands a good sequence alignment, and working with thousands of sequences is quite a challenge. However, there are very efficient tools and filtering parameters to overcome this, as we will try and clarify in this response. But first, we would like to point out that codeml does have some limitations. Some of our dS values appear as “0”, when in fact it is not zero, but a truncated integer with more than six decimal places. In codeml, a dS = 0 results in an infinite dN/dS represented by 999 (as explained in the frequently asked questions of PAML manual available at: <http://abacus.gene.ucl.ac.uk/software/pamlFAQs.pdf>). These were filtered out by us. Hence, all dN/dS ratios were calculated with a dS higher than 0. While revising our table, we observed that, after converting it from txt to xlsx, some values were also mistakenly replaced by “0”. We corrected these errors in lines: 2-8, 10, 11, 13, 17-19, 21, 23, 25-27, 29-31, 33, 36 and 38-40. The revised table (now Supplementary Table 8) is submitted with this reviewed version. We apologize for this and thank the reviewer for the careful perusal of these results.

The reviewer is right that low values of dS and great values of dN/dS could indicate problems in our alignments. Here, we used three strategies to overcome this problem: 1) the alignments were done with the amino acid sequences and then back-translated into a nucleotide alignment, which increases accuracy (Bininda-Emonds 2005; Abascal et al. 2010); 2) gaps and poorly aligned regions were masked with gblocks (Talavera and Castresana 2007); 3) we avoided the infinite dN/dS values and filtered dS exactly equal to 0 (dN/dS = infinity). We revised all 40 alignments and we did observe that some blocks are more variable but none of them seems to be a result of alignment errors.

The reviewer is also correct that the LRT test tells us which model fits better our dN/dS estimations. We tested two models, one in which the dN/dS was fixed and equal 1 that we considered as a null model and another where the dN/dS is free to be estimated as proposed by PAML author (Yang 2007). When the p-value is > 0.05, the hypothesis of a neutral model (dN/dS = 1) is not rejected. On the other hand, if the p-value is < 0.05 the null hypothesis is rejected, meaning that there might be a selective pressure on the compared sequence. When we “accept” the alternative hypothesis, we use the dN/dS ratio to evaluate whether the selective pressure is to decreased variation (dN/dS < 1, purifying selection) or fixed mutations (dN/dS > 1, positive selection). We reviewed the manuscript and did not find any use of the p-values given by PAML to conclude whether a particular gene is submitted to positive selection or not, but we clarified how we reached this conclusion by adding the following sentences in the methods section “The LRT test indicates which model fits better our d_N/d_S estimations. Thus α determines which genes are evolving under neutrality (null hypothesis) or a selective regime. If neutrality is rejected, then d_N/d_S values indicate purifying (d_N/d_S < 1) or positive selection (d_N/d_S > 1,)”

*The last suggestion raised by the reviewer was the use of a branch model using datamonkey. We did use a lineage-based analysis contrasting the *C. hominivorax* branch against the whole tree in codeml.*

There other models implemented in datamonkey such as some branch-sites models there are more sensitive to find positive selection as dN/dS can vary among each site. These branch-site tests are also implemented in PAML. Datamonkey is not really an approach itself; it is a web-based interface to a server that allows you test your data with many different algorithms. Datamonkey is indeed very interesting, but the analysis can only be done for a single gene (sampled over multiple taxa). We did this for 4,476 genes, and the analysis was easily automated with a batch script. It is not possible to use Datamonkey for genome-wide analysis. It is possible, however, to download and run HyPhy locally, but there is no reason to assume it would be better than PAML.

Branch-site models to detect episodic diversifying selection on single amino-acid sites, does increase the power of the test, but at the same time, this model is also more sensitive to alignment errors (Yang and dos Reis 2011) and more difficult to report and interpret in genome-wide analysis. With our tests, we are able to detect selection only if the dN/dS ratio averaged over all sites is greater than one. This is a very conservative analysis (Yang 2007), but it is less biased by alignment errors. We think it is the most appropriate strategy for our genome-wide exploratory analysis.

References

- Abascal F, Zardoya R, Telford MJ (2010) TranslatorX: multiple alignment of nucleotide sequences guided by amino acid translations. *Nucleic Acids Res* 38:W7–W13. doi: 10.1093/nar/gkq291
- Bininda-Emonds ORP (2005) transAlign: using amino acids to facilitate the multiple alignment of protein-coding DNA sequences. *BMC Bioinformatics* 6:156. doi: 10.1186/1471-2105-6-156
- Talavera G, Castresana J (2007) Improvement of Phylogenies after Removing Divergent and Ambiguously Aligned Blocks from Protein Sequence Alignments. *Syst Biol* 56:564–577. doi: 10.1080/10635150701472164
- Yang Z (2007) PAML 4: Phylogenetic Analysis by Maximum Likelihood. *Mol Biol Evol* 24:1586–1591. doi: 10.1093/molbev/msm088
- Yang Z, dos Reis M (2011) Statistical Properties of the Branch-Site Test of Positive Selection. *Mol Biol Evol* 28:1217–1228. doi: 10.1093/molbev/msq303

3. I do not understand why the authors chose to illustrate the probability (on the x axis of A) that a model of selection is better than another for each gene. This does not give a sense of whether a gene is under strong positive selection or not (please see my comment above). This figure is also extremely hard to read and interpret.

We have revised the figure (Fig. 6). The reviewer is correct that the p -values do not inform if the gene is under strong selection or not. It informs that we rejected the null model of

neutral evolution ($dN/dS = 1$), and the results is better explained by selection (positive or purifying). We have edited this figure to improve simplicity and indicate the dN/dS ratios (dot size). The larger the dot, the higher the positive selection onto the protein (averaged over all codons). This shows the 40 genes under positive selection but at varying levels. The p -value (color scale) show that the null hypothesis of neutral evolution was rejected for these genes.

4. Figure 7. It is unclear to me what the Y axis of the graph illustrates (expression relative to what value ?)

In the “Developmental Expression Analyses” section of the paper we noted that RNAseq reads were mapped to genes and expression values converted to transcript per million mapped (TPM). We have now added to the legend for Figure 7 that the normalized values are in transcript per million mapped.

5. I could not see the right part of Figure 1 on the pdf version of the manuscript and wasn't able to identify the individual file in which this figure might have been additionally submitted. In addition, in the legend of this figure, there might be a problem with the following sentence : « Benchmarking Universal Single-Copy Orthologs [17] analyses based on the dipteran dataset (odb8). »

We have edited the Figure 1 legend to improve clarity as suggested. Part B of Figure 1 has been corrected.

6. Is the phylogenetic position of *C. hominivorax* obtained in this study expected compared to previous analyses?

Yes, the placement is as expected. We have noted in the text and added a citation on dipteran phylogeny.

7. Line 586-588 : « Indeed, clusters of U6 snRNA genes were found in two contigs that showed extensive similarity and thus are likely from different alleles rather than a duplication. »

This sentence describes a result. Thus it would better fit in the result/discussion section where the authors first compare the size of their assembly to the expected size of the genome.

Sentence deleted as the result had been stated earlier in the section on U6 genes (Supp Fig 14).

8. I here list a number of typos that I have spotted but it is very likely that there are many more as I have not read the entire manuscript with an homogeneous level of effort.

All of the minor corrections listed below have been made. We appreciate the list provided by this referee.

Line 160 : please correct « suggests »

Line 195 : replace « in number in » by « in number to » ?

Line 130 : Please correct « Our goal of these analyses »

Line 141 : Please correct « aminoglygan »

Line 242 : the repetition of « associated » does not look great

Line 252 : perhaps replace « the highest expression in levels embryos and male » by « the highest expression in embryos and males » ?

Figure 4 is overall very hard to read

Legend of Fig 5 : replace « categorizes » with « categories » ?

Line 333 : « Developmental RNA-seq two gene ontology categories under positive selection based on additional table 16. »

There seems to be a problem with this sentence.

Line 345 : replace « following » by « followed » ?

Line 385 : « males » instead of « male »

Line 390 : please correct the sentence

Line 448 : please correct « propapotic »

Line 584 : Do you mean 94 Mb ?

REVIEWERS' COMMENTS:

Reviewer #1 (Remarks to the Author):

This study will give a new scope for comparative genomic research for parasitic fly species. Additionally, because majority of forensically important fly species belong to the same family, Calliphoridae, this study can be a valuable reference for forensic entomogenetics. I think the revised version is acceptable.

Reviewer #2 (Remarks to the Author):

I confirmed that the authors revised the figures and text according to my comments. I have a minor comment about the legend for Fig. 5. The explanation of asterisks (significant by Mann-Whitney U-test) has been removed but should be returned to the legend for clarity. In panel A, it seems curious that two GO terms (nucleus, sequence-specific DNA binding transcription factor activity) have significantly different dN/dS despite their medians coincide with the genome-wide median. The use of Mann-Whitney U-test for comparison between partial (by GO-term) and global medians (or means) may not be appropriate because the distribution of values can differ greatly between them and can return significant results even when the means are the same. The median dN/dS of individual GO term coincides with the genome-wide median for many GO-terms, and other medians often take the same value as if they are categorized. Just curious how could these happen, though I may have misunderstood something. I hope the authors will make appropriate changes for the above concerns if necessary.

Reviewer #3 (Remarks to the Author):

Thank you for addressing my comments

Comment by the reviewer: "In panel A, it seems curious that two GO terms (nucleus, sequence-specific DNA binding transcription factor activity) have significantly different dN/dS despite their medians coincide with the genome-wide median. The use of Mann-Whitney U-test for comparison between partial (by GO-term) and global medians (or means) may not be appropriate because the distribution of values can differ greatly between them and can return significant results even when the means are the same. The median dN/dS of individual GO term coincides with the genome-wide median for many GO-terms, and other medians often take the same value as if they are categorized. Just curious how could these happen, though I may have misunderstood something."

Response by the authors: It is indeed curious to find a significant difference for that groups with equal medians, but it is not uncommon, because the Mann-Whitney U-tests (MWU) are rank sum tests and not median tests [1].

If the compared groups have an identical or similar distributions and have identical medians, then we expect no significant difference between them. However, if the medians are equal, but the distributions are different, we can indeed find a significant difference.

The MWU test is based on the rank values obtained by combining the two samples. This is done by ordering these values, from the smallest to the largest, regardless of the fact from which group each value comes from. The sums of the ranks from one group is then compared with the expected rank sum. With sufficient sample sizes, the difference in ranks will be large enough to be significant, even with equal medians.

In the examples highlighted by the reviewer, we do have equal means, but the shape of the distribution, and therefore, the rank sums, are different. And this is exactly what we want to test: whether the distributions of dN/dS are identical in these different groups of genes (without assuming them to follow the normal distribution).

The MWU is the most appropriate test for our comparisons due to the large differences in the sample sizes when comparing genes in a given GO category vs remaining genes in the genome. This test is specifically designed to deal with this unbalanced sampling [1]. The test is used for many other comparisons of smaller groups of genes vs. genomes, as shown in the examples below, just to name a few:

- comparison of the sequence divergence of genes in the X-chromosome with genes in the autosomes of *Drosophila* [2];
- comparison of selenoprotein genes to other genes that regulate Se in aquatic vertebrates [3];
- comparison of the rate of protein evolution (dN/dS) of single copy genes with duplicated genes in primates [4];

- comparison of dN/dS ratios among genes within different GO categories in Drosophila (similar to our approach) [5].

Changes to manuscript:

-We modified a sentence in the Figure 4 legend to be clearer on the statistical analysis. The sentence is “*, represents values with a significant positive correlation based on a regression-based p-value for assessing the statistical significance ($P < 0.05$) between a stage and module⁹¹.”

-In the “Statistics and Reproducibility” section of methods we added: “Statistic analyses were performed in the R software environment. The package ‘dplyr’ was used for large data manipulation and the package ‘ggplot2’ was used for creating graphics. Mann-Whitney tests were done with the ‘wilcox.test’ function of the core R package. The significance level, alpha, was set to 0.05, and false discovery rate correction was done for multiple tests. Other specific tests used in this study are described in the relevant section (e.g. comparative genomics)”

[1] Mann HB, Whitney DR (1947) On a Test of Whether one of Two Random Variables is Stochastically Larger than the Other. *Ann. Math. Statist.*, 18(1), 50-60.

[2] Coolon JD, Stevenson KR, McManus CJ, Yang B, Graveley BR, Wittkopp PJ(2015) Molecular Mechanisms and Evolutionary Processes Contributing to Accelerated Divergence of Gene Expression on the Drosophila X Chromosome. *Mol. Biol. Evol.*, 32(10):2605–2615.

[3] Sarangi GK, Romagné F, Castellano S (2018) Distinct Patterns of Selection in Selenium-Dependent Genes between Land and Aquatic Vertebrates. *Mol. Biol. Evol.*, 35 (7): 1744–1756.

[4] O’Toole AN, Hurst LD, McLysaght A (2018) Faster Evolving Primate Genes Are More Likely to Duplicate. *Mol. Biol. Evol.*, 35(1): 107–118.

[5] Drosophila 12 Genomes Consortium (2007) Evolution of genes and genomes on the Drosophila phylogeny. *Nature*, 450: 203–218.